# Sample Complexity of Tree Search Configuration: Cutting Planes and Beyond

**Maria-Florina Balcan**
School of Computer Science
Carnegie Mellon University
`ninamf@cs.cmu.edu`

**Siddharth Prasad**
Computer Science Department
Carnegie Mellon University
`sprasad2@cs.cmu.edu`

**Tuomas Sandholm**
Computer Science Department
Carnegie Mellon University
Optimized Markets, Inc.
Strategic Machine, Inc.
Strategy Robot, Inc.
`sandholm@cs.cmu.edu`

**Ellen Vitercik**
EECS Department
UC Berkeley
`vitercik@berkeley.edu`

## Abstract

Cutting-plane methods have enabled remarkable successes in integer programming over the last few decades. State-of-the-art solvers integrate a myriad of cutting-plane techniques to speed up the underlying tree-search algorithm used to find optimal solutions. In this paper we provide sample complexity bounds for cut-selection in branch-and-cut (B&C). Given a training set of integer programs sampled from an application-specific input distribution and a family of cut selection policies, these guarantees bound the number of samples sufficient to ensure that using any policy in the family, the size of the tree B&C builds on average over the training set is close to the expected size of the tree B&C builds. We first bound the sample complexity of learning cutting planes from the canonical family of Chvátal-Gomory cuts. Our bounds handle any number of waves of any number of cuts and are fine tuned to the magnitudes of the constraint coefficients. Next, we prove sample complexity bounds for more sophisticated cut selection policies that use a combination of scoring rules to choose from a family of cuts. Finally, beyond the realm of cutting planes for integer programming, we develop a general abstraction of tree search that captures key components such as node selection and variable selection. For this abstraction, we bound the sample complexity of learning a good policy for building the search tree.

## 1 Introduction

Integer programming is one of the most broadly-applicable tools in computer science, used to formulate problems from operations research (such as routing, scheduling, and pricing), machine learning (such as adversarially-robust learning, MAP estimation, and clustering), and beyond. *Branch-and-cut (B&C)* is the most widely-used algorithm for solving integer programs (IPs). B&C is highly configurable, and with a deft configuration, it can be used to solve computationally challenging problems. Finding a good configuration, however, is a notoriously difficult problem.

We study machine learning approaches to configuring policies for selecting *cutting planes*, which have an enormous impact on B&C's performance. At a high level, B&C works by recursively partitioning the IP's feasible region, searching for the locally optimal solution within each set of the partition,

35th Conference on Neural Information Processing Systems (NeurIPS 2021).

until it can verify that it has found the globally optimal solution. An IP's feasible region is defined by a set of linear inequalities $A\boldsymbol{x} \leq \boldsymbol{b}$ and integer constraints $\boldsymbol{x} \in \mathbb{Z}^n$, where $n$ is the number of variables. By dropping the integrality constraints, we obtain the *linear programming (LP) relaxation* of the IP, which can be solved efficiently. A cutting plane is a carefully-chosen linear inequality $\boldsymbol{\alpha}^T \boldsymbol{x} \leq \beta$ which refines the LP relaxation's feasible region without separating any integral point. Intuitively, a well-chosen cutting plane will remove a large portion of the LP relaxation's feasible region, speeding up the time it takes B&C to find the optimal solution to the original IP. Cutting plane selection is a crucial task, yet it is challenging because many cutting planes and cut-selection policies have tunable parameters, and the best configuration depends intimately on the application domain.

We provide the first provable guarantees for learning high-performing cutting planes and cut-selection policies, tailored to the application at hand. We model the application domain via an unknown, application-specific distribution over IPs, as is standard in the literature on using machine learning for integer programming [e.g., 21, 23, 31, 36, 43]. For example, this could be a distribution over the routing IPs that a shipping company must solve day after day. The learning algorithm's input is a training set sampled from this distribution. The goal is to use this training set to learn cutting planes and cut-selection policies with strong future performance on problems from the same application but which are not already in the training set—or more formally, strong expected performance.

## 1.1 Summary of main contributions and overview of techniques

As our first main contribution, we provide *sample complexity bounds* of the following form: fixing a family of cutting planes, we bound the number of samples sufficient to ensure that for any sequence of cutting planes from the family, the average size of the B&C tree is close to the expected size of the B&C tree. We measure performance in terms of the size of the search tree B&C builds. Our guarantees apply to the parameterized family of *Chvátal-Gomory (CG) cuts* [10, 17], one of the most widely-used families of cutting planes.

The overriding challenge is that to provide guarantees, we must analyze how the tree size changes as a function of the cut parameters. This is a sensitive function—slightly shifting the parameters can cause the tree size to shift from constant to exponential in the number of variables. Our key technical insight is that as the parameters vary, the entries of the cut (i.e., the vector $\boldsymbol{\alpha}$ and offset $\beta$ of the cut $\boldsymbol{\alpha}^T \boldsymbol{x} \leq \beta$) are multivariate polynomials of bounded degree. The number of terms defining the polynomials is exponential in the number of parameters, but we show that the polynomials can be embedded in a space with dimension sublinear in the number of parameters. This insight allows us to better understand tree size as a function of the parameters. We then leverage results by Balcan et al. [8] that show how to use structure exhibited by dual functions (measuring an algorithm's performance, such as its tree size, as a function of its parameters) to derive sample complexity bounds.

Our second main contribution is a sample complexity bound for learning cut-selection policies, which allow B&C to adaptively select cuts as it solves the input IP. These cut-selection policies assign a number of real-valued scores to a set of cutting planes and then apply the cut that has the maximum weighted sum of scores. Tree size is a volatile function of these weights, though we prove that it is piecewise constant, as illustrated in Figure 1, which allows us to prove our sample complexity bound.

Finally, as our third main contribution, we provide guarantees for tuning weighted combinations of scoring rules for other aspects of tree search beyond cut selection, including node and variable selection. We prove that there is a set of hyperplanes splitting the parameter space into regions such that if tree search uses any configuration from a single region, it will take the same sequence of actions. This structure allows us to prove our sample complexity bound. This is the first paper to provide guarantees for tree search configuration that apply simultaneously to multiple different aspects of the algorithm—prior research was specific to variable selection [5].

Sample complexity bounds are important because if the parameterized class of cuts or cut-selection policies that we optimize over is highly complex and the training set is too small, the learned cut or cut-selection policy might have great average empirical performance over the training set but terrible future performance. In other words, the parameter configuration procedure may overfit to the training set. The sample complexity bounds we provide are uniform-convergence: we prove that given enough samples, uniformly across all parameter settings, the difference between average and empirical performance is small. In other words, these bounds hold for any procedure one might use to optimize over the training set: manual or automated, optimal or suboptimal. No matter what

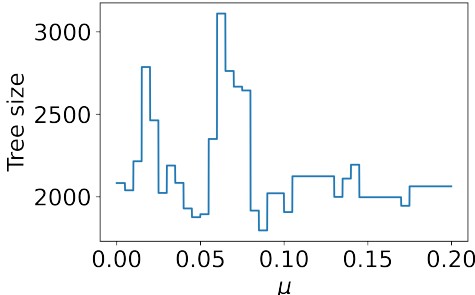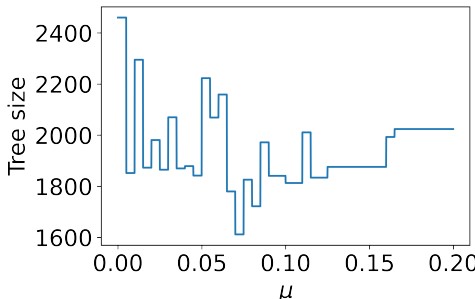

Figure 1: Two examples of tree size as a function of a SCIP cut-selection parameter $\mu$ (the directed cutoff distance weight, defined in Section 2) on IPs generated from the Combinatorial Auctions Test Suite [30] (the "regions" generator with 600 bids and 600 goods). SCIP [16] is the leading open-source IP solver.

parameter setting the configuration procedure comes up with, the user can be guaranteed that so long as that parameter setting has good average empirical performance over the training set, it will also have strong future performance.

## 1.2 Related work

**Applied research on tree search configuration.** Over the past decade, a substantial literature has developed on the use of machine learning for integer programming and tree search [e.g., 2, 7, 9, 13, 19, 23–25, 29, 31–33, 35, 36, 41–43]. This has included research that improves specific aspects of B&C such as variable selection [2, 13, 24, 29, 32, 41], node selection [19, 35, 44], and heuristic scheduling [25]. These papers are applied, whereas we focus on providing theoretical guarantees.

With respect to cutting plane selection, the focus of this paper, Sandholm [36] uses machine learning techniques to customize B&C for combinatorial auction winner determination, including cutting plane selection. Tang et al. [37] and Huang et al. [20] study machine learning approaches to cutting plane selection. The former work formulates this problem as a reinforcement learning problem and shows that their approach can outperform human-designed heuristics for a variety of tasks. The latter work studies cutting plane selection in the multiple-instance-learning framework and proposes a neural-network architecture for scoring and ranking cutting planes. Meanwhile, the focus of our paper is to provide the first provable guarantees for cutting plane selection via machine learning.

Ferber et al. [15] study a problem where the IP objective vector $c$ is unknown, but an estimate $\hat{c}$ can be obtained from data. Their goal is to optimize the quality of the solutions obtained by solving the IP defined by $\hat{c}$, with respect to the true vector $c$. They do so by formulating the IP as a differentiable layer in a neural network. The nonconvex nature of the IP does not allow for straightforward gradient computation for the backward pass, so they obtain a continuous surrogate using cutting planes.

**Provable guarantees for algorithm configuration.** Gupta and Roughgarden [18] initiated the study of sample complexity bounds for algorithm configuration. In research most related to ours, Balcan et al. [5] provide sample complexity bounds for learning tree search *variable selection policies (VSPs)*. They prove their bounds by showing that for any IP, hyperplanes partition the VSP parameter space into regions where the B&C tree size is a constant function of the parameters. The analysis in this paper requires new techniques because although we prove that the B&C tree size is a piecewise-constant function of the CG cutting plane parameters, the boundaries between pieces are far more complex than hyperplanes: they are hypersurfaces defined by multivariate polynomials.

Kleinberg et al. [26, 27] and Weisz et al. [38, 39] design configuration procedures for runtime minimization that come with theoretical guarantees. Their algorithms are designed for the case where there are a finitely-many parameter settings to choose from (although they are still able to provide guarantees for infinite parameter spaces by running their procedure on a finite sample of configurations; Balcan et al. [5, 6] analyze when discretization approaches can and cannot be gainfully employed). In contrast, our guarantees are designed for infinite parameter spaces.

## 2 Problem formulation

In this section we give a more detailed technical overview of branch-and-cut, as well as an overview of the tools from learning theory we use to prove sample complexity guarantees.

### 2.1 Branch-and-cut

We study integer programs (IPs) in canonical form given by

$$\max \left\{ \boldsymbol{c}^T \boldsymbol{x} : A\boldsymbol{x} \leq \boldsymbol{b}, \boldsymbol{x} \geq 0, \boldsymbol{x} \in \mathbb{Z}^n \right\}, \tag{1}$$

where $A \in \mathbb{Z}^{m \times n}$, $\boldsymbol{b} \in \mathbb{Z}^m$, and $\boldsymbol{c} \in \mathbb{R}^n$. Branch-and-cut (B&C) works by recursively partitioning the input IP's feasible region, searching for the locally optimal solution within each set of the partition until it can verify that it has found the globally optimal solution. It organizes this partition as a search tree, with the input IP stored at the root. It begins by solving the LP relaxation of the input IP; we denote the solution as $\boldsymbol{x}_{\mathsf{LP}}^* \in \mathbb{R}^n$. If $\boldsymbol{x}_{\mathsf{LP}}^*$ satisfies the IP's integrality constraints ($\boldsymbol{x}_{\mathsf{LP}}^* \in \mathbb{Z}^n$), then the procedure terminates—$\boldsymbol{x}_{\mathsf{LP}}^*$ is the globally optimal solution. Otherwise, it uses a *variable selection policy* to choose a variable $x[i]$. In the left child of the root, it stores the original IP with the additional constraint that $x[i] \leq \lfloor x_{\mathsf{LP}}^*[i] \rfloor$, and in the right child, with the additional constraint that $x[i] \geq \lceil x_{\mathsf{LP}}^*[i] \rceil$. It then uses a *node selection policy* to select a leaf of the tree and repeats this procedure—solving the LP relaxation and branching on a variable. B&C can *fathom* a node, meaning that it will stop searching along that branch, if 1) the LP relaxation satisfies the IP's integrality constraints, 2) the LP relaxation is infeasible, or 3) the objective value of the LP relaxation's solution is no better than the best integral solution found thus far. We assume there is a bound $\kappa$ on the size of the tree we allow B&C to build before we terminate, as is common in prior research [5, 21, 26, 27].

Cutting planes are a means of ensuring that at each iteration of B&C, the solution to the LP relaxation is as close to the optimal integral solution as possible. Formally, let $\mathcal{P} = \{ \boldsymbol{x} \in \mathbb{R}^n : A\boldsymbol{x} \leq \boldsymbol{b}, \boldsymbol{x} \geq 0 \}$ denote the feasible region obtained by taking the LP relaxation of IP (1). Let $\mathcal{P}_I = \mathrm{conv}(\mathcal{P} \cap \mathbb{Z}^n)$ denote the integer hull of $\mathcal{P}$. A *valid* cutting plane is any hyperplane $\boldsymbol{\alpha}^T \boldsymbol{x} \leq \beta$ such that if $\boldsymbol{x}$ is in the integer hull ($\boldsymbol{x} \in \mathcal{P}_I$), then $\boldsymbol{x}$ satisfies the inequality $\boldsymbol{\alpha}^T \boldsymbol{x} \leq \beta$. In other words, a valid cut does not remove any integral point from the LP relaxation's feasible region. A valid cutting plane *separates* $\boldsymbol{x} \in \mathcal{P} \setminus \mathcal{P}_I$ if it does not satisfy the inequality, or in other words, $\boldsymbol{\alpha}^T \boldsymbol{x} > \beta$. At any node of the search tree, B&C can add valid cutting planes that separate the optimal solution to the node's LP relaxation, thus improving the solution estimates used to prune the search tree. However, adding too many cuts will increase the time it takes to solve the LP relaxation at each node. Therefore, solvers such as SCIP [16], the leading open-source solver, bound the number of cuts that will be applied.

A famous class of cutting planes is the family of *Chvátal-Gomory (CG) cuts*[1] [10, 17], which are parameterized by vectors $\boldsymbol{u} \in \mathbb{R}^m$. The CG cut defined by $\boldsymbol{u} \in \mathbb{R}^m$ is the hyperplane $\lfloor \boldsymbol{u}^T A \rfloor \boldsymbol{x} \leq \lfloor \boldsymbol{u}^T \boldsymbol{b} \rfloor$, which is guaranteed to be valid. Throughout this paper we primarily restrict our attention to $\boldsymbol{u} \in [0, 1)^m$. This is without loss of generality, since the facets of $\mathcal{P} \cap \{ \boldsymbol{x} \in \mathbb{R}^n : \lfloor \boldsymbol{u}^T A \rfloor \boldsymbol{x} \leq \lfloor \boldsymbol{u}^T \boldsymbol{b} \rfloor \, \forall \boldsymbol{u} \in \mathbb{R}^m \}$ can be described by the finitely many $\boldsymbol{u} \in [0, 1)^m$ such that $\boldsymbol{u}^T A \in \mathbb{Z}^n$ (Lemma 5.13 of Conforti et al. [11]).

Some IP solvers such as SCIP use *scoring rules* to select among cutting planes, which are meant to measure the quality of a cut. Some commonly-used scoring rules include *efficacy* [4] ($\mathtt{score}_1$), *objective parallelism* [1] ($\mathtt{score}_2$), *directed cutoff distance* [16] ($\mathtt{score}_3$), and *integral support* [40] ($\mathtt{score}_4$) (defined in Appendix A). *Efficacy* measures the distance between the cut $\boldsymbol{\alpha}^T \boldsymbol{x} \leq \beta$ and $\boldsymbol{x}_{\mathsf{LP}}^*$: $\mathtt{score}_1(\boldsymbol{\alpha}^T \boldsymbol{x} \leq \beta) = (\boldsymbol{\alpha}^T \boldsymbol{x}_{\mathsf{LP}}^* - \beta) / \|\boldsymbol{\alpha}\|_2$, as illustrated in Figure 2a. *Objective parallelism* measures the angle between the objective $\boldsymbol{c}$ and the cut's normal vector $\boldsymbol{\alpha}$: $\mathtt{score}_2(\boldsymbol{\alpha}^T \boldsymbol{x} \leq \beta) = |\boldsymbol{c}^T \boldsymbol{\alpha}| / (\|\boldsymbol{\alpha}\|_2 \|\boldsymbol{c}\|_2)$, as illustrated in Figures 2b and 2c. *Directed cutoff distance* measures the distance between the LP optimal solution and the cut in a more relevant direction than the efficacy scoring rule. Specifically, let $\overline{\boldsymbol{x}}$ be the *incumbent solution*, which is the best-known feasible solution to the input IP. The directed cutoff distance is the distance between the hyperplane $(\boldsymbol{\alpha}, \beta)$ and the current LP solution $\boldsymbol{x}_{\mathsf{LP}}^*$ along the direction of the incumbent $\overline{\boldsymbol{x}}$, as illustrated in Figures 2d and 2e: $\mathtt{score}_3(\boldsymbol{\alpha}^T \boldsymbol{x} \leq \beta) = \|\overline{\boldsymbol{x}} - \boldsymbol{x}_{\mathsf{LP}}^*\|_2 \cdot (\boldsymbol{\alpha}^T \boldsymbol{x}_{\mathsf{LP}}^* - \beta) / |\boldsymbol{\alpha}^T (\overline{\boldsymbol{x}} - \boldsymbol{x}_{\mathsf{LP}}^*)|$. SCIP uses the scoring rule $\frac{3}{5}\mathtt{score}_1 + \frac{1}{10}\mathtt{score}_2 + \frac{1}{2}\mathtt{score}_3 + \frac{1}{10}\mathtt{score}_4$ [16].

---

[1]The set of CG cuts is equivalent to the set of Gomory (fractional) cuts [12], another commonly studied family of cutting planes with a slightly different parameterization.

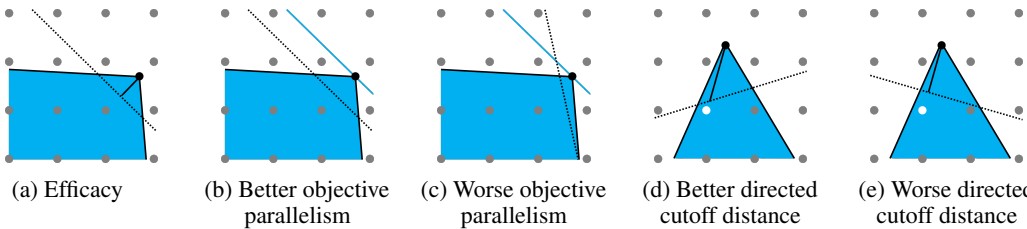

| (a) Efficacy | (b) Better objective parallelism | (c) Worse objective parallelism | (d) Better directed cutoff distance | (e) Worse directed cutoff distance |

Figure 2: Illustration of scoring rules. In each figure, the blue region is the feasible region, the black dotted line is the cut in question, the blue solid line is orthogonal to the objective $c$, the black dot is the LP optimal solution, and the white dot is the incumbent IP solution. Figure 2a illustrates efficacy, which is the length of the black solid line between the cut and the LP optimal solution. The cut in Figure 2b has better objective parallelism than the cut in Figure 2c. The cut in Figure 2d has a better directed cutoff distance than the cut in Figure 2e, but both have the same efficacy.

## 2.2 Learning theory background and notation

The goal of this paper is to learn cut-selection policies using samples in order to guarantee, with high probability, that B&C builds a small tree in expectation on unseen IPs. To this end, we rely on the notion of *pseudo-dimension* [34], a well-known measure of a function class's *intrinsic complexity*. The pseudo-dimension of a function class $\mathcal{F} \subseteq \mathbb{R}^{\mathcal{Y}}$, denoted $\mathrm{Pdim}(\mathcal{F})$, is the largest integer $N$ for which there exist $N$ inputs $y_1, \ldots, y_N \in \mathcal{Y}$ and $N$ thresholds $r_1, \ldots, r_N \in \mathbb{R}$ such that for every $(\sigma_1, \ldots, \sigma_N) \in \{0, 1\}^N$, there exists $f \in \mathcal{F}$ such that $f(y_i) \geq r_i$ if and only if $\sigma_i = 1$. Function classes with bounded pseudo-dimension satisfy the following uniform convergence guarantee [3, 34]. Let $[-\kappa, \kappa]$ be the range of the functions in $\mathcal{F}$, let $N_{\mathcal{F}}(\varepsilon, \delta) = O(\frac{\kappa^2}{\varepsilon^2}(\mathrm{Pdim}(\mathcal{F}) + \ln(\frac{1}{\delta})))$, and let $N \geq N_{\mathcal{F}}(\varepsilon, \delta)$. For all distributions $\mathcal{D}$ on $\mathcal{Y}$, with probability $1 - \delta$ over the draw of $y_1, \ldots, y_N \sim \mathcal{D}$, for every function $f \in \mathcal{F}$, the average value of $f$ over the samples is within $\varepsilon$ of its expected value: $|\frac{1}{N} \sum_{i=1}^{N} f(y_i) - \mathbb{E}_{y \sim \mathcal{D}}[f(y)]| \leq \varepsilon$. The quantity $N_{\mathcal{F}}(\varepsilon, \delta)$ is the *sample complexity* of $\mathcal{F}$.

We use the notation $\|A\|_{1,1}$ to denote the sum of the absolute values of all the entries in $A$.

## 3 Learning Chvátal-Gomory cuts

In this section we bound the sample complexity of learning CG cuts at the root node of the B&C search tree. In many IP settings, similar IPs are being solved and there can be good cuts that carry across instances—for example, in applications where the constraints stay the same or roughly the same across instances,[2] and only the objective changes. One high-stakes example of this is the feasibility checking problem in the billion-dollar incentive auction for radio spectrum, where prices change but the radiowave interference constraints do not change.

We warm up by analyzing the case where a single CG cut is added at the root (Section 3.1), and then build on this analysis to handle $w$ sequential *waves* of $k$ simultaneous CG cuts (Section 3.3). This means that all $k$ cuts in the first wave are added simultaneously, the new (larger) LP relaxation is solved, all $k$ cuts in the second wave are added to the new problem simultaneously, and so on. B&C adds cuts in waves because otherwise, the angles between cuts would become obtuse, leading to numerical instability. Moreover, many commercial IP solvers only add cuts at the root because those cuts can be leveraged throughout the tree. However, in Section 5, we also provide guarantees for applying cuts throughout the tree. In this section, we assume that all aspects of B&C (such as node selection and variable selection) are fixed except for the cuts applied at the root of the search tree.

### 3.1 Learning a single cut

To provide sample complexity bounds, as per Section 2.2, we bound the pseudo-dimension of the set of functions $f_{\boldsymbol{u}}$ for $\boldsymbol{u} \in [0, 1]^m$, where $f_{\boldsymbol{u}}(\boldsymbol{c}, A, \boldsymbol{b})$ is the size of the tree B&C builds when it applies the CG cut defined by $\boldsymbol{u}$ at the root. To do so, we take advantage of structure exhibited by the class of *dual* functions, each of which is defined by a fixed IP $(\boldsymbol{c}, A, \boldsymbol{b})$ and measures tree size as

---

[2]We assume that constraints are generated in the same order across instances; see Appendix B for a discussion.

a function of the parameters $\boldsymbol{u}$. In other words, each dual function $f_{\boldsymbol{c},A,\boldsymbol{b}}^* : [0,1]^m \to \mathbb{R}$ is defined as $f_{\boldsymbol{c},A,\boldsymbol{b}}^*(\boldsymbol{u}) = f_{\boldsymbol{u}}(\boldsymbol{c}, A, \boldsymbol{b})$. Our main result in this section is a proof that the dual functions are well-structured (Lemma 3.2), which then allows us to apply a result by Balcan et al. [8] to bound $\text{Pdim}(\{f_{\boldsymbol{u}} : \boldsymbol{u} \in [0,1]^m\})$ (Theorem 3.3). Proving that the dual functions are well-structured is challenging because they are volatile: slightly perturbing $\boldsymbol{u}$ can cause the tree size to shift from constant to exponential in $n$, as we prove in the following theorem. The full proof is in Appendix C.

**Theorem 3.1.** *For any integer $n$, there exists an integer program $(\boldsymbol{c}, A, \boldsymbol{b})$ with two constraints and $n$ variables such that if $\frac{1}{2} \le u[1] - u[2] < \frac{n+1}{2n}$, then applying the CG cut defined by $\boldsymbol{u}$ at the root causes B&C to terminate immediately. Meanwhile, if $\frac{n+1}{2n} \le u[1] - u[2] < 1$, then applying the CG cut defined by $\boldsymbol{u}$ at the root causes B&C to build a tree of size at least $2^{(n-1)/2}$.*

*Proof sketch.* Without loss of generality, assume that $n$ is odd. Consider an IP with constraints $2(x[1] + \cdots + x[n]) \le n$, $-2(x[1] + \cdots + x[n]) \le -n$, $\boldsymbol{x} \in \{0,1\}^n$, and any objective. This IP is infeasible because $n$ is odd. Jeroslow [22] proved that without the use of cutting planes or heuristics, B&C will build a tree of size $2^{(n-1)/2}$ before it terminates. We prove that when $\frac{1}{2} \le u[1] - u[2] < \frac{n+1}{2n}$, the CG cut halfspace defined by $\boldsymbol{u} = (u[1], u[2])$ has an empty intersection with the feasible region of the IP, causing B&C to terminate immediately. On the other hand, we show that if $\frac{n+1}{2n} \le u[1] - u[2] < 1$, then the CG cut halfspace defined by $\boldsymbol{u}$ contains the feasible region of the IP, and thus leaves the feasible region unchanged. In this case, due to Jeroslow [22], applying this CG cut at the root will cause B&C to build a tree of size at least $2^{(n-1)/2}$ before it terminates. $\square$

This theorem shows that the dual tree-size functions can be extremely sensitive to perturbations in the CG cut parameters. However, we are able to prove that the dual functions are piecewise-constant.

**Lemma 3.2.** *For any IP $(\boldsymbol{c}, A, \boldsymbol{b})$, there are $O(\|A\|_{1,1} + \|\boldsymbol{b}\|_1 + n)$ hyperplanes that partition $[0,1]^m$ into regions where in any one region $R$, the dual function $f_{\boldsymbol{c},A,\boldsymbol{b}}^*(\boldsymbol{u})$ is constant for all $\boldsymbol{u} \in R$.*

*Proof.* Let $\boldsymbol{a}_1, \ldots, \boldsymbol{a}_n \in \mathbb{R}^m$ be the columns of $A$. Let $A_i = \|\boldsymbol{a}_i\|_1$ and $B = \|\boldsymbol{b}\|_1$, so for any $\boldsymbol{u} \in [0,1]^m$, $\lfloor \boldsymbol{u}^T \boldsymbol{a}_i \rfloor \in [-A_i, A_i]$ and $\lfloor \boldsymbol{u}^T \boldsymbol{b} \rfloor \in [-B, B]$. For each integer $k_i \in [-A_i, A_i]$, we have $\lfloor \boldsymbol{u}^T \boldsymbol{a}_i \rfloor = k_i \iff k_i \le \boldsymbol{u}^T \boldsymbol{a}_i < k_i + 1$. There are $\sum_{i=1}^n 2A_i + 1 = O(\|A\|_{1,1} + n)$ such halfspaces, plus an additional $2B + 1$ halfspaces of the form $k_{n+1} \le \boldsymbol{u}^T \boldsymbol{b} < k_{n+1} + 1$ for each $k_{n+1} \in \{-B, \ldots, B\}$. In any region $R$ defined by the intersection of these halfspaces, the vector $(\lfloor \boldsymbol{u}^T \boldsymbol{a}_1 \rfloor, \ldots, \lfloor \boldsymbol{u}^T \boldsymbol{a}_n \rfloor, \lfloor \boldsymbol{u}^T \boldsymbol{b} \rfloor)$ is constant for all $\boldsymbol{u} \in R$, and thus so is the resulting cut. $\square$

Combined with the main result of Balcan et al. [8], this lemma implies the following bound.

**Theorem 3.3.** *Let $\mathcal{F}_{\alpha,\beta}$ denote the set of all functions $f_{\boldsymbol{u}}$ for $\boldsymbol{u} \in [0,1]^m$ defined on the domain of IPs $(\boldsymbol{c}, A, \boldsymbol{b})$ with $\|A\|_{1,1} \le \alpha$ and $\|\boldsymbol{b}\|_1 \le \beta$. Then, $\text{Pdim}(\mathcal{F}_{\alpha,\beta}) = O(m \log(m(\alpha + \beta + n)))$.*

This theorem implies that $\widetilde{O}(\kappa^2 m / \varepsilon^2)$ samples are sufficient to ensure that with high probability, for every CG cut, the average size of the tree B&C builds upon applying the cutting plane is within $\epsilon$ of the expected size of the tree it builds (the $\widetilde{O}$ notation suppresses logarithmic terms).

### 3.2 Learning a sequence of cuts

We now determine the sample complexity of making $w$ sequential CG cuts at the root. The first cut is defined by $m$ parameters $\boldsymbol{u}_1 \in [0,1]^m$ for each of the $m$ constraints. Its application leads to the addition of the row $\lfloor \boldsymbol{u}_1^T A \rfloor \boldsymbol{x} \le \lfloor \boldsymbol{u}_1^T \boldsymbol{b} \rfloor$ to the constraint matrix. The next cut is then be defined by $m + 1$ parameters $\boldsymbol{u}_2 \in [0,1]^{m+1}$ since there are now $m + 1$ constraints. Continuing in this fashion, the $w$th cut is be defined by $m + w - 1$ parameters $\boldsymbol{u}_w \in [0,1]^{m+w-1}$. Let $f_{\boldsymbol{u}_1,\ldots,\boldsymbol{u}_w}(\boldsymbol{c}, A, \boldsymbol{b})$ be the size of the tree B&C builds when it applies the CG cut defined by $\boldsymbol{u}_1$, then applies the CG cut defined by $\boldsymbol{u}_2$ to the new IP, and so on, all at the root of the search tree.

As in Section 3.1, we bound the pseudo-dimension of the functions $f_{\boldsymbol{u}_1,\ldots,\boldsymbol{u}_w}$ by analyzing the structure of the dual functions $f_{\boldsymbol{c},A,\boldsymbol{b}}^*$, which measure tree size as a function of the parameters $\boldsymbol{u}_1, \ldots, \boldsymbol{u}_w$. Specifically, $f_{\boldsymbol{c},A,\boldsymbol{b}}^* : [0,1]^m \times \cdots \times [0,1]^{m+w-1} \to \mathbb{R}$, where $f_{\boldsymbol{c},A,\boldsymbol{b}}^*(\boldsymbol{u}_1, \ldots, \boldsymbol{u}_w) = f_{\boldsymbol{u}_1,\ldots,\boldsymbol{u}_w}(\boldsymbol{c}, A, \boldsymbol{b})$. The analysis in this section is more complex because the $s^{th}$ cut (with $s \in$

$\{2, \ldots, W\}$) depends not only on the parameters $\boldsymbol{u}_s$ but also on $\boldsymbol{u}_1, \ldots, \boldsymbol{u}_{s-1}$. We prove that the dual functions are again piecewise-constant, but in this case, the boundaries between pieces are defined by multivariate polynomials rather than hyperplanes. The full proof is in Appendix C.

**Lemma 3.4.** *For any IP* $(\boldsymbol{c}, A, \boldsymbol{b})$*, there are* $O(w2^w \|A\|_{1,1} + 2^w \|\boldsymbol{b}\|_1 + nw)$ *multivariate polynomials in* $\leq w^2 + mw$ *variables of degree* $\leq w$ *that partition* $[0,1]^m \times \cdots \times [0,1]^{m+w-1}$ *into regions where in any one region* $R$*,* $f^*_{\boldsymbol{c},A,\boldsymbol{b}}(\boldsymbol{u}_1, \ldots, \boldsymbol{u}_w)$ *is constant for all* $(\boldsymbol{u}_1, \ldots, \boldsymbol{u}_w) \in R$.

*Proof sketch.* Let $\boldsymbol{a}_1, \ldots, \boldsymbol{a}_n \in \mathbb{R}^m$ be the columns of $A$. For $\boldsymbol{u}_1 \in [0,1]^m, \ldots, \boldsymbol{u}_w \in [0,1]^{m+w-1}$, define $\widetilde{\boldsymbol{a}}_i^1 \in [0,1]^m, \ldots, \widetilde{\boldsymbol{a}}_i^w \in [0,1]^{m+w-1}$ for each $i \in [n]$ such that $\widetilde{\boldsymbol{a}}_i^s$ is the $i$th column of the constraint matrix after applying cuts $\boldsymbol{u}_1, \ldots, \boldsymbol{u}_{s-1}$. Similarly, define $\widetilde{\boldsymbol{b}}^s$ to be the constraint vector after applying the first $s-1$ cuts. More precisely, we have the recurrence relation

$$\widetilde{\boldsymbol{a}}_i^1 = \boldsymbol{a}_i \qquad\qquad \widetilde{\boldsymbol{b}}^1 = \boldsymbol{b}$$

$$\widetilde{\boldsymbol{a}}_i^s = \begin{bmatrix} \widetilde{\boldsymbol{a}}_i^{s-1} \\ \boldsymbol{u}_{s-1}^T \widetilde{\boldsymbol{a}}_i^{s-1} \end{bmatrix} \qquad \widetilde{\boldsymbol{b}}^s = \begin{bmatrix} \widetilde{\boldsymbol{b}}^{s-1} \\ \boldsymbol{u}_{s-1}^T \widetilde{\boldsymbol{b}}^{s-1} \end{bmatrix}$$

for $s = 2, \ldots, W$. We prove that $\lfloor \boldsymbol{u}_s^T \widetilde{\boldsymbol{a}}_i^s \rfloor \in [-2^{s-1}\|\boldsymbol{a}_i\|_1, 2^{s-1}\|\boldsymbol{a}_i\|_1]$. For each integer $k_i$ in this interval, $\lfloor \boldsymbol{u}_s^T \widetilde{\boldsymbol{a}}_i^s \rfloor = k_i \iff k_i \leq \boldsymbol{u}_s^T \widetilde{\boldsymbol{a}}_i^s < k_i + 1$. The boundaries of these surfaces are defined by polynomials over $\boldsymbol{u}_s$ in $\leq ms + s^2$ variables with degree $\leq s$. Counting the total number of such hypersurfaces yields the lemma statement. $\qquad\square$

We now use this structure to provide a pseudo-dimension bound. The full proof is in Appendix C.

**Theorem 3.5.** *Let* $\mathcal{F}_{\alpha,\beta}$ *denote the set of all functions* $f_{\boldsymbol{u}_1, \ldots, \boldsymbol{u}_w}$ *for* $\boldsymbol{u}_1 \in [0,1]^m, \ldots, \boldsymbol{u}_w \in [0,1]^{m+w-1}$ *defined on the domain of integer programs* $(\boldsymbol{c}, A, \boldsymbol{b})$ *with* $\|A\|_{1,1} \leq \alpha$ *and* $\|\boldsymbol{b}\|_1 \leq \beta$. *Then,* $\mathrm{Pdim}(\mathcal{F}_{\alpha,\beta}) = O(mw^2 \log(mw(\alpha + \beta + n)))$.

*Proof sketch.* The space of $0/1$ classifiers induced by the set of degree $\leq w$ multivariate polynomials in $w^2 + mw$ variables has VC dimension $O((w^2 + mw)\log w)$ [3]. However, we more carefully examine the structure of the polynomials considered in Lemma 3.4 to give an improved VC dimension bound of $1 + mw$. For each $j = 1, \ldots, m$ define $\widetilde{\boldsymbol{u}}_1[j], \ldots, \widetilde{\boldsymbol{u}}_w[j]$ recursively as

$$\widetilde{\boldsymbol{u}}_1[j] = \boldsymbol{u}_1[j]$$

$$\widetilde{\boldsymbol{u}}_s[j] = \boldsymbol{u}_s[j] + \sum_{\ell=1}^{s-1} \boldsymbol{u}_s[m+\ell]\widetilde{\boldsymbol{u}}_\ell[j] \qquad \text{for } s = 2, \ldots, w$$

The space of polynomials induced by the $s$th cut is contained in $\mathrm{span}\{1, \widetilde{\boldsymbol{u}}_s[1], \ldots, \widetilde{\boldsymbol{u}}_s[m]\}$. The intuition for this is as follows: consider the additional term added by the $s$th cut to the constraint matrix, that is, $\boldsymbol{u}_s^T \widetilde{\boldsymbol{a}}_i^s$. The first $m$ coordinates $(\boldsymbol{u}_s[1], \ldots, \boldsymbol{u}_s[m])$ interact only with $\boldsymbol{a}_i$—so $\boldsymbol{u}_s[j]$ collects a coefficient of $\boldsymbol{a}_i[j]$. Each subsequent coordinate $\boldsymbol{u}_s[m+\ell]$ interacts with all coordinates of $\widetilde{\boldsymbol{a}}_i^s$ arising from the first $\ell$ cuts. The term that collects a coefficient of $\boldsymbol{a}_i[j]$ is precisely $\boldsymbol{u}_s[m+\ell]$ times the sum of all terms from the first $\ell$ cuts with a coefficient of $\boldsymbol{a}_i[j]$. Using standard facts about the VC dimension of vector spaces and their duals in conjunction with Lemma 3.4 and the framework of Balcan et al. [8] yields the theorem statement. $\qquad\square$

The sample complexity (defined in Section 2.2) of learning $W$ sequential cuts is thus $\widetilde{O}(\kappa^2 mw^2/\epsilon^2)$.

### 3.3 Learning waves of simultaneous cuts

We now determine the sample complexity of making $w$ sequential waves of cuts at the root, each wave consisting of $k$ simultaneous CG cuts. Given vectors $\boldsymbol{u}_1^1, \ldots, \boldsymbol{u}_1^k \in [0,1]^m, \boldsymbol{u}_2^1, \ldots, \boldsymbol{u}_2^k \in [0,1]^{m+k}, \ldots, \boldsymbol{u}_w^1, \ldots, \boldsymbol{u}_w^k \in [0,1]^{m+k(w-1)}$, let $f_{\boldsymbol{u}_1^1, \ldots, \boldsymbol{u}_1^k, \ldots, \boldsymbol{u}_w^1, \ldots, \boldsymbol{u}_w^k}(\boldsymbol{c}, A, \boldsymbol{b})$ be the size of the tree B&C builds when it applies the CG cuts defined by $\boldsymbol{u}_1^1, \ldots, \boldsymbol{u}_1^k$, then applies the CG cuts defined by $\boldsymbol{u}_2^1, \ldots, \boldsymbol{u}_2^k$ to the new IP, and so on, all at the root of the search tree. The full proof of the following theorem is in Appendix C, and follows from the observation that $w$ waves of $k$ simultaneous cuts can be viewed as making $kw$ sequential cuts with the restriction that cuts within each wave assign nonzero weight only to constraints from previous waves.

**Theorem 3.6.** *Let $\mathcal{F}_{\alpha,\beta}$ be the set of all functions $f_{\boldsymbol{u}_1^1,\ldots,\boldsymbol{u}_1^k,\ldots,\boldsymbol{u}_w^1,\ldots,\boldsymbol{u}_w^k}$ for $\boldsymbol{u}_1^1,\ldots,\boldsymbol{u}_1^k \in [0,1]^m,\ldots,\boldsymbol{u}_w^1,\ldots,\boldsymbol{u}_w^k \in [0,1]^{m+k(w-1)}$ defined on the domain of integer programs $(\boldsymbol{c}, A, \boldsymbol{b})$ with $\|A\|_{1,1} \leq \alpha$ and $\|\boldsymbol{b}\|_1 \leq \beta$. Then, $\mathrm{Pdim}(\mathcal{F}_{\alpha,\beta}) = O(mk^2w^2 \log(mkw(\alpha + \beta + n)))$.*

This result implies that the sample complexity of learning $W$ waves of $k$ cuts is $\widetilde{O}(\kappa^2 mk^2w^2/\epsilon^2)$.

### 3.4 Data-dependent guarantees

So far, our guarantees have depended on the maximum possible norms of the constraint matrix and vector in the domain of IPs under consideration. The uniform convergence result in Section 2.2 for $\mathcal{F}_{\alpha,\beta}$ only holds for distributions over $A$ and $\boldsymbol{b}$ with norms bounded by $\alpha$ and $\beta$, respectively. In Appendix C.1, we show how to convert these into more broadly applicable data-dependent guarantees that leverage properties of the distribution over IPs. These guarantees hold without assumptions on the distribution's support, and depend on $\mathbb{E}[\max_i \|A_i\|_{1,1}]$ and $\mathbb{E}[\max_i \|\boldsymbol{b}_i\|_1]$ (where the expectation is over $N$ samples), thus giving a sharper sample complexity guarantee that is tuned to the distribution.

## 4 Learning cut selection policies

In Section 3, we studied the sample complexity of learning waves of specific cut parameters. In this section, we bound the sample complexity of learning *cut-selection policies* at the root, that is, functions that take as input an IP and output a candidate cut. Using scoring rules is a more nuanced way of choosing cuts since it allows for the cut parameters to depend on the input IP.

Formally, let $\mathcal{I}_m$ be the set of IPs with $m$ constraints (the number of variables is always fixed at $n$) and let $\mathcal{H}_m$ be the set of all hyperplanes in $\mathbb{R}^m$. A *scoring rule* is a function $\texttt{score} : \cup_m(\mathcal{H}_m \times \mathcal{I}_m) \to \mathbb{R}_{\geq 0}$. The real value $\texttt{score}(\boldsymbol{\alpha}^T\boldsymbol{x} \leq \beta, (\boldsymbol{c}, A, \boldsymbol{b}))$ is a measure of the quality of the cutting plane $\boldsymbol{\alpha}^T\boldsymbol{x} \leq \beta$ for the IP $(\boldsymbol{c}, A, \boldsymbol{b})$. Examples include the scoring rules discussed in Section 2.1.

Suppose $\texttt{score}_1, \ldots, \texttt{score}_d$ are $d$ different scoring rules. We now bound the sample complexity of learning a combination of these scoring rules that guarantee a low expected tree size. Our high-level proof technique is the same as in the previous section: we establish that the dual tree-size functions are piecewise structured, and then apply the general framework of Balcan et al. [8] to obtain pseudo-dimension bounds.

**Theorem 4.1.** *Let $\mathcal{C}$ be a set of cutting-plane parameters such that for every IP $(\boldsymbol{c}, A, \boldsymbol{b})$, there is a decomposition of $\mathcal{C}$ into $\leq r$ regions such that the cuts generated by any two vectors in the same region are the same. Let $\texttt{score}_1, \ldots, \texttt{score}_d$ be $d$ scoring rules. For $\boldsymbol{\mu} \in \mathbb{R}^d$, let $f_{\boldsymbol{\mu}}(\boldsymbol{c}, A, \boldsymbol{b})$ be the size of the tree B&C builds when it chooses a cut from $\mathcal{C}$ to maximize $\mu[1]\texttt{score}_1(\cdot, (\boldsymbol{c}, A, \boldsymbol{b})) + \cdots + \mu[d]\texttt{score}_d(\cdot, (\boldsymbol{c}, A, \boldsymbol{b}))$. Then, $\mathrm{Pdim}(\{f_{\boldsymbol{\mu}} : \boldsymbol{\mu} \in \mathbb{R}^d\}) = O(d\log(rd))$.*

*Proof.* Fix an integer program $(\boldsymbol{c}, A, \boldsymbol{b})$. Let $\boldsymbol{u}_1, \ldots, \boldsymbol{u}_r \in \mathcal{C}$ be representative cut parameters for each of the $r$ regions. Consider the hyperplanes $\sum_{i=1}^d \mu[i]\texttt{score}_i(\boldsymbol{u}_s) = \sum_{i=1}^d \mu[i]\texttt{score}_i(\boldsymbol{u}_t)$ for each $s \neq t \in \{1, \ldots, r\}$ (suppressing the dependence on $\boldsymbol{c}, A, \boldsymbol{b}$). These $O(r^2)$ hyperplanes partition $\mathbb{R}^d$ into regions such that as $\boldsymbol{\mu}$ varies in a given region, the cut chosen from $\mathcal{C}$ is invariant. The desired pseudo-dimension bound follows from the main result of Balcan et al. [8]. $\square$

Theorem 4.1 can be directly instantiated with the class of CG cuts. Combining Lemma 3.2 with the basic combinatorial fact that $k$ hyperplanes partition $\mathbb{R}^m$ into at most $k^m$ regions, we get that the pseudo-dimension of $\{f_{\boldsymbol{\mu}} : \boldsymbol{\mu} \in \mathbb{R}^d\}$ defined on IPs with $\|A\|_{1,1} \leq \alpha$ and $\|\boldsymbol{b}\|_1 \leq \beta$ is $O(dm\log(d(\alpha + \beta + n)))$. Instantiating Theorem 4.1 with the set of all sequences of $w$ CG cuts requires the following extension of scoring rules to sequences of cutting planes. A *sequential scoring rule* is a function that takes as input an IP $(\boldsymbol{c}, A, \boldsymbol{b})$ and a sequence of cutting planes $h_1, \ldots, h_w$, where each cut lives in one higher dimension than the previous. It measures the quality of this sequence of cutting planes when applied one after the other to the original IP. Every scoring rule $\texttt{score}$ can be naturally extended to a sequential scoring rule $\overline{\texttt{score}}$ defined by $\overline{\texttt{score}}(h_1, \ldots, h_w, (\boldsymbol{c}^0, A^0, \boldsymbol{b}^0)) = \sum_{i=0}^{w-1} \texttt{score}(h_{i+1}, (\boldsymbol{c}^i, A^i, \boldsymbol{b}^i))$, where $(\boldsymbol{c}^i, A^i, \boldsymbol{b}^i)$ is the IP after applying cuts $h_1, \ldots, h_{i-1}$.

**Corollary 4.2.** *Let $\mathcal{C} = [0,1]^m \times \cdots \times [0,1]^{m+w-1}$ denote the set of possible sequences of $w$ Chvátal-Gomory cut parameters. Let $\texttt{score}_1, \ldots, \texttt{score}_d : \mathcal{C} \times \mathcal{I}_m \times \cdots \times \mathcal{I}_{m+w-1} \to \mathbb{R}$*

be $d$ sequential scoring rules and let $f_{\boldsymbol{\mu}}(\boldsymbol{c}, A, \boldsymbol{b})$ be as in Theorem 4.1 for the class $\mathcal{C}$. Then, $\text{Pdim}(\{f_{\boldsymbol{\mu}}^w : \boldsymbol{\mu} \in \mathbb{R}^d\}) = O(dmw^2 \log(dw(\alpha + \beta + n)))$.

*Proof.* In Lemma 3.4 and Theorem 3.5 we showed that there are $O(w2^w\alpha + 2^w\beta + nw)$ multivariate polynomials that belong to a family of polynomials $\mathcal{G}$ with $\text{VCdim}(\mathcal{G}^*) \leq 1 + mw$ ($\mathcal{G}^*$ denotes the dual of $\mathcal{G}$) that partition $\mathcal{C}$ into regions such that resulting sequence of cuts is invariant in each region. By Claim 3.5 by Balcan et al. [8], the number of regions is $O(w2^w\alpha + 2^w\beta + nw)^{\text{VCdim}(\mathcal{G}^*)} \leq O(w2^w\alpha + 2^w\beta + nw)^{1+mw}$. The corollary then follows from Theorem 4.1. $\qquad\square$

These results bound the sample complexity of learning cut-selection policies based on scoring rules, which allow the cuts B&C that selects to depend on the input IP.

## 5 Sample complexity of generic tree search

In this section, we study the sample complexity of selecting high-performing parameters for generic tree-based algorithms, which are a generalization of B&C. This abstraction allows us to provide guarantees for simultaneously optimizing key aspects of tree search beyond cut selection, including node selection and branching variable selection. We also generalize the previous sections by allowing actions (such as cut selection) to be taken at any stage of the tree search—not just at the root.

Tree search algorithms take place over a series of $\kappa$ *rounds* (analogous to the B&B tree-size cap $\kappa$ in the previous sections). There is a sequence of $t$ *steps* that the algorithm takes on each round. For example, in B&C, these steps include node selection, cut selection, and variable selection. The specific *action* the algorithm takes during each step (for example, which node to select, which cut to include, or which variable to branch on) typically depends on a *scoring rule*. This scoring rule weights each possible action and the algorithm performs the action with the highest weight. These actions (deterministically) transition the algorithm from one *state* to another. This high-level description of tree search is summarized by Algorithm 1. For each step $j \in [t]$, the number of possible actions is $T_j \in \mathbb{N}$. There is a scoring rule $\texttt{score}_j$, where $\texttt{score}_j(k, s) \in \mathbb{R}$ is the weight associated with the action $k \in [T_j]$ when the algorithm is in the state $s$.

---

**Algorithm 1** Tree search

**Input:** Problem instance, $t$ scoring rules $\texttt{score}_1, \ldots, \texttt{score}_t$, number of rounds $\kappa$.
1: $s_{1,1} \leftarrow$ Initial state of algorithm
2: **for** each round $i \in [\kappa]$ **do**
3:     **for** each step $j \in [t]$ **do**
4:         Perform the action $k \in [T_j]$ that maximizes $\texttt{score}_j(s_{i,j}, k)$
5:         $s_{i,j+1} \leftarrow$ New state of algorithm
6:     $s_{i+1,1} \leftarrow s_{i,t+1}$         ▷ State at beginning of next round equals state at end of this round
**Output:** Incumbent solution in state $s_{\kappa,t+1}$, if one exists.

---

There are often several scoring rules one could use, and it is not clear which to use in which scenarios. As in Section 4, we provide guarantees for learning combinations of these scoring rules for the particular application at hand. More formally, for each step $j \in [t]$, rather than just a single scoring rule $\texttt{score}_j$ as in Step 4, there are $d_j$ scoring rules $\texttt{score}_{j,1}, \ldots, \texttt{score}_{j,d_j}$. Given parameters $\boldsymbol{\mu}_j = (\mu_j[1], \ldots, \mu_j[d_j]) \in \mathbb{R}^{d_j}$, the algorithm takes the action $k \in [T_j]$ that maximizes $\sum_{i=1}^{d_j} \mu_j[i]\texttt{score}_{j,i}(k, s)$. There is a distribution $\mathcal{D}$ over inputs $x$ to Algorithm 1. For example, when this framework is instantiated for branch-and-cut, $x$ is an integer program $(\boldsymbol{c}, A, \boldsymbol{b})$. There is a utility function $f_{\boldsymbol{\mu}}(x) \in [-H, H]$ that measures the utility of the algorithm parameterized by $\boldsymbol{\mu} = (\boldsymbol{\mu}_1, \ldots, \boldsymbol{\mu}_t)$ on input $x$. For example, this utility function might measure the size of the search tree that the algorithm builds. We assume that this utility function is *final-state-constant*:

**Definition 5.1.** Let $\boldsymbol{\mu} = (\boldsymbol{\mu}_1, \ldots, \boldsymbol{\mu}_t)$ and $\boldsymbol{\mu}' = (\boldsymbol{\mu}_1', \ldots, \boldsymbol{\mu}_t')$ be two parameter vectors. Suppose that we run Algorithm 1 on input $x$ once using the scoring rule $\texttt{score}_j = \sum_{i=1}^{d_j} \mu_j[i]\texttt{score}_{j,i}$ and once using the scoring rule $\texttt{score}_j = \sum_{i=1}^{d_j} \mu_j'[i]\texttt{score}_{j,i}$. Suppose that on each run, we obtain the same final state $s_{\kappa,t+1}$. The utility function is *final-state-constant* if $f_{\boldsymbol{\mu}}(x) = f_{\boldsymbol{\mu}'}(x)$.

We provide a sample complexity bound for learning the parameters $\boldsymbol{\mu}$. The full proof is in Appendix D.

**Theorem 5.2.** *Let $d = \sum_{j=1}^{t} d_j$ denote the total number of tunable parameters of tree search. Then,*

$$\text{Pdim}\left(\left\{f_{\boldsymbol{\mu}} : \boldsymbol{\mu} \in \mathbb{R}^d\right\}\right) = O\left(d\kappa \sum_{j=1}^{t} \log T_j + d \log d\right).$$

*Proof sketch.* We prove that there is a set of hyperplanes splitting the parameter space into regions such that if tree search uses any parameter setting from a single region, it will always take the same sequence of actions (including node, variable, and cut selection). The main subtlety is an induction argument to count these hyperplanes that depends on the current step of the tree-search algorithm. □

In the context of integer programming, Theorem 5.2 not only recovers the main result of Balcan et al. [5] for learning variable selection policies, but also yields a more general bound that simultaneously incorporates cutting plane selection, variable selection, and node selection. In B&C, the first action of each round is to select a node. Since there are at most $2^{n+1} - 1$ nodes, $T_1 \leq 2^{n+1} - 1$. The second action is to choose a cutting plane. As in Theorem 4.1, let $\mathcal{C}$ be a family of cutting planes such that for every IP $(\boldsymbol{c}, A, \boldsymbol{b})$, there is a decomposition of the parameter space into $\leq r$ regions such that the cuts generated by any two parameters in the same region are the same. So $T_2 \leq r$. The last action is to choose a variable to branch on at that node, so $T_3 = n$. Applying Theorem 5.2, $\text{Pdim}(\{f_{\boldsymbol{\mu}} : \boldsymbol{\mu} \in \mathbb{R}^d\}) = O(d\kappa n + d\kappa \log r + d \log d)$. Ignoring $T_1$ and $T_2$, thereby only learning the variable selection policy, recovers the $O(d\kappa \log n + d \log d)$ bound of Balcan et al. [5].

## 6   Conclusions and future research

We provided the first provable guarantees for using machine learning to configure cutting planes and cut-selection policies. We analyzed the sample complexity of learning cutting planes from the popular family of Chvátal-Gomory (CG) cuts. We then provided sample complexity guarantees for learning parameterized cut-selection policies, which allow the branch-and-cut algorithm to adaptively apply cuts as it builds the search tree. We showed that this analysis can be generalized to simultaneously capture various key aspects of tree search beyond cut selection, such as node and variable selection.

This paper opens up a variety questions for future research. For example, which other cut families can we learn over with low sample complexity? Section 3 focused on learning within the family of CG cuts (Sections 4 and 5 applied more generally). There are many other families, such as *Gomory mixed-integer cuts* and *lift-and-project cuts*, and a sample complexity analysis of these is an interesting direction for future research (and would call for new techniques). In addition, can we use machine learning to design improved scoring rules and heuristics for cut selection? The bounds we provide in Section 4 apply to any choice of scoring rules, no matter how simple or complex. Is it possible to obtain even better bounds by taking into account the complexity of the scoring rules? Finally, the bounds in this paper are worst case, but a great direction for future research is to develop data-dependent bounds that improve based on the structure of the input distribution.

## Acknowledgements

This material is based on work supported by the National Science Foundation under grants IIS-1618714, IIS-1718457, IIS-1901403, CCF-1733556, CCF-1535967, CCF-1910321, SES-1919453, the ARO under award W911NF2010081, DARPA under cooperative agreement HR00112020003, an AWS Machine Learning Research Award, an Amazon Research Award, a Bloomberg Research Grant, and a Microsoft Research Faculty Fellowship.

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
