## A  Additional background about cutting planes

**Integral support [40].**  Let $Z$ be the set of all indices $\ell \in [n]$ such that $\boldsymbol{\alpha}[\ell] \neq 0$. Let $\bar{Z}$ be the set of all indices $\ell \in Z$ such that the $\ell^{th}$ variable is constrained to be integral. This scoring rule is defined as

$$\texttt{score}_4(\boldsymbol{\alpha}^T \boldsymbol{x} \leq \beta) = \frac{|\bar{Z}|}{|Z|}.$$

Wesselmann and Suhl [40] write that "one may argue that a cut having non-zero coefficients on many (possibly fractional) integer variables is preferable to a cut which consists mostly of continuous variables."

## B  Constraint ordering

Integer programs are typically automatically generated and the generation code typically generates constraints in the same order for all instances. More formally, if we fix an ordering over the variables, we can simply assume without loss of generality that the constraints and cuts are ordered lexicographically, so the constraints cannot be permuted across instances. (In a bit more detail, given constraints with coefficients $[a_1, \ldots, a_n, b]$ and $[a'_1, \ldots, a'_n, b']$, the first constraint would come first in the lexicographic order if $a_1 > a'_1$ and second if $a_1 < a'_1$. If $a_1 = a'_1$ it would come first in the order if $a_2 > a'_2$ and second if $a_2 < a'_2$, and so on.)

## C  Omitted results and proofs from Section 3

*Proof of Theorem 3.1.* Without loss of generality, we assume that $n$ is odd. We define the integer program

$$\begin{array}{ll} \text{maximize} & 0 \\ \text{subject to} & 2x[1] + \cdots + 2x[n] = n \\ & \boldsymbol{x} \in \{0, 1\}^n, \end{array} \quad (2)$$

which is infeasible because $n$ is odd. Jeroslow [22] proved that without the use of cutting planes or heuristics, B&C will build a tree of size $2^{(n-1)/2}$ before it terminates. Rewriting the equality constraint as $2x[1] + \cdots + 2x[n] \leq n$ and $-2\left(x[1] + \cdots + x[n]\right) \leq -n$, a CG cut defined by the vector $\boldsymbol{u} \in \mathbb{R}^2_{\geq 0}$ will have the form $\lfloor 2(u[1] - u[2]) \rfloor \left(x[1] + \cdots + x[n]\right) \leq \lfloor n\left(u[1] - u[2]\right) \rfloor.$

Suppose that $\frac{1}{2} \leq u[1] - u[2] < \frac{n+1}{2n}$. On the left-hand-side of the constraint, $\lfloor 2(u[1] - u[2]) \rfloor = 1$. On the right-hand-side of the constraint, $n\left(u[1] - u[2]\right) < \frac{n+1}{2}$. Since $n$ is odd, $\frac{n+1}{2}$ is an integer, which means that $\lfloor n\left(u[1] - u[2]\right) \rfloor \leq \frac{n-1}{2}$. Therefore, the CG cut defined by $\boldsymbol{u}$ satisfies the inequality $x[1] + \cdots + x[n] \leq \frac{n-1}{2}$, as illustrated in Figure 3a. The intersection of this halfspace with the feasible region of the original integer program (Equation (2)) is empty, so applying this CG cut at the root will cause B&C to terminate immediately.

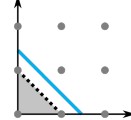

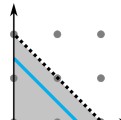

(a) Cut produced when $\frac{1}{2} \leq u[1] - u[2] < \frac{2}{3}$. The grey solid region is the set of points $\boldsymbol{x}$ such that $x[1] + x[2] \leq 1$.

(b) Cut produced when $\frac{2}{3} \leq u[1] - u[2] < 1$. The grey solid region is the set of points $\boldsymbol{x}$ such that $x[1] + x[2] \leq 2$.

Figure 3: Illustration of Theorem 3.1 when $n = 3$, projected onto the $x[3] = 0$ plane. The blue solid line is the feasible region $2x[1] + 2x[2] = 3$. The black dotted lines are the cut.

Meanwhile, suppose that $\frac{n+1}{2n} \leq u[1] - u[2] < 1$. Then it is still the case that $\lfloor 2(u[1] - u[2]) \rfloor = 1$. Also, $n\left(u[1] - u[2]\right) \geq \frac{n+1}{2}$, which means that $\lfloor n\left(u[1] - u[2]\right) \rfloor \geq \frac{n+1}{2}$. Therefore, the CG cut defined by $\boldsymbol{u}$ dominates the inequality $x[1] + \cdots + x[n] \leq \frac{n+1}{2}$, as illustrated in Figure 3b. The intersection of this halfspace with the feasible region of the original integer program is equal to the

integer program's feasible region, so by Jeroslow's result [22], applying this CG cut at the root will cause B&C to build a tree of size at least $2^{(n-1)/2}$ before it terminates. $\qquad\square$

*Proof of Lemma 3.4.* Let $\boldsymbol{a}_1, \ldots, \boldsymbol{a}_n \in \mathbb{R}^m$ be the columns of $A$. For $\boldsymbol{u}_1 \in [0,1]^m, \ldots, \boldsymbol{u}_w \in [0,1]^{m+w-1}$, define $\widetilde{\boldsymbol{a}}_i^1 \in [0,1]^m, \ldots, \widetilde{\boldsymbol{a}}_i^w \in [0,1]^{m+w-1}$ for each $i = 1, \ldots, n$ such that $\widetilde{\boldsymbol{a}}_i^s$ is the $i$th column of the constraint matrix after applying cuts $\boldsymbol{u}_1, \ldots, \boldsymbol{u}_{s-1}$. More precisely, $\widetilde{\boldsymbol{a}}_i^1 \in [0,1]^m, \ldots, \widetilde{\boldsymbol{a}}_i^w \in [0,1]^{m+w-1}$ are defined recursively as

$$\widetilde{\boldsymbol{a}}_i^1 = \boldsymbol{a}_i$$
$$\widetilde{\boldsymbol{a}}_i^s = \begin{bmatrix} \widetilde{\boldsymbol{a}}_i^{s-1} \\ \boldsymbol{u}_{s-1}^T \widetilde{\boldsymbol{a}}_i^{s-1} \end{bmatrix}$$

for $s = 2, \ldots, w$. Similarly, define $\widetilde{\boldsymbol{b}}^s$ to be the constraint vector after applying the first $s-1$ cuts:

$$\widetilde{\boldsymbol{b}}^1 = \boldsymbol{b}$$
$$\widetilde{\boldsymbol{b}}^s = \begin{bmatrix} \widetilde{\boldsymbol{b}}^{s-1} \\ \boldsymbol{u}_{s-1}^T \widetilde{\boldsymbol{b}}^{s-1} \end{bmatrix}$$

for $s = 2, \ldots, w$. (These vectors depend on the cut vectors, but we will suppress this dependence for the sake of readability).

We prove this lemma by showing that there are $O(w2^w \|A\|_{1,1} + 2^w \|\boldsymbol{b}\|_1 + nw)$ hypersurfaces determined by polynomials that partition $[0,1]^m \times \cdots \times [0,1]^{m+w-1}$ into regions where in any one region $R$, the $w$ cuts

$$\sum_{i=1}^n \lfloor \boldsymbol{u}_1^T \widetilde{\boldsymbol{a}}_i^1 \rfloor x[i] \le \left\lfloor \boldsymbol{u}_1^T \widetilde{\boldsymbol{b}}^1 \right\rfloor$$

$$\vdots$$

$$\sum_{i=1}^n \lfloor \boldsymbol{u}_w^T \widetilde{\boldsymbol{a}}_i^w \rfloor x[i] \le \left\lfloor \boldsymbol{u}_w^T \widetilde{\boldsymbol{b}}^w \right\rfloor$$

are invariant across all $(\boldsymbol{u}_1, \ldots, \boldsymbol{u}_w) \in R$. To this end, let $A_i = \|\boldsymbol{a}_i\|_1$ and $B = \|\boldsymbol{b}\|_1$. For each $s \in \{1, \ldots, w\}$, we claim that

$$\lfloor \boldsymbol{u}_s^T \widetilde{\boldsymbol{a}}_i^s \rfloor \in \left[ -2^{s-1} A_i, 2^{s-1} A_i \right].$$

We prove this by induction. The base case of $s = 1$ is immediate since $\widetilde{\boldsymbol{a}}_i^1 = \boldsymbol{a}_i$ and $\boldsymbol{u} \in [0,1]^m$. Suppose now that the claim holds for $s$. By the induction hypothesis,

$$\left\| \widetilde{\boldsymbol{a}}_i^{s+1} \right\|_1 = \left\| \begin{bmatrix} \widetilde{\boldsymbol{a}}_i^s \\ \boldsymbol{u}_s^T \widetilde{\boldsymbol{a}}_i^s \end{bmatrix} \right\|_1 = \| \widetilde{\boldsymbol{a}}_i^s \|_1 + |\boldsymbol{u}_s^T \widetilde{\boldsymbol{a}}_i^s| \le 2 \| \widetilde{\boldsymbol{a}}_i^s \|_1 \le 2^s A_i,$$

so

$$\lfloor \boldsymbol{u}_{s+1}^T \widetilde{\boldsymbol{a}}_i^{s+1} \rfloor \in \left[ -\left\| \widetilde{\boldsymbol{a}}_i^{s+1} \right\|_1, \left\| \widetilde{\boldsymbol{a}}_i^{s+1} \right\|_1 \right] \subseteq [-2^s A_i, 2^s A_i],$$

as desired. Now, for each integer $k_i \in [-2^{s-1} A_i, 2^{s-1} A_i]$, we have

$$\lfloor \boldsymbol{u}_s^T \widetilde{\boldsymbol{a}}_i^s \rfloor = k_i \iff k_i \le \boldsymbol{u}_s^T \widetilde{\boldsymbol{a}}_i^s < k_i + 1.$$

$\boldsymbol{u}_s^T \widetilde{\boldsymbol{a}}_i^s$ is a polynomial in variables $\boldsymbol{u}_1[1], \ldots, \boldsymbol{u}_1[m], \boldsymbol{u}_2[1], \ldots, \boldsymbol{u}_2[m+1], \ldots, \boldsymbol{u}_s[1], \ldots, \boldsymbol{u}_s[m+s-1]$, for a total of $\le ms + s^2$ variables. Its degree is at most $s$. There are thus a total of

$$\sum_{s=1}^w \sum_{i=1}^n (2 \cdot 2^{s-1} A_i + 1) = O\left( w 2^w \|A\|_{1,1} + nw \right)$$

polynomials each of degree at most $w$ plus an additional $\sum_{s=1}^w (2 \cdot 2^{s-1} B + 1) = O(2^w B + w)$ polynomials of degree at most $w$ corresponding to the hypersurfaces of the form

$$k_{n+1} \le \boldsymbol{u}_s^T \widetilde{\boldsymbol{b}}^s < k_{n+1} + 1$$

for each $s$ and each $k_{n+1} \in \{-2^{s-1} B, \ldots, 2^{s-1} B\}$. This yields a total of $O(w 2^w \|A\|_{1,1} + 2^w \|\boldsymbol{b}\|_1 + nw)$ polynomials in $\le mw + w^2$ variables of degree $\le w$. $\qquad\square$

*Proof of Theorem 3.5.* The space of polynomials induced by the $s$th cut, that is, $\{k + \boldsymbol{u}_s^T \widetilde{\boldsymbol{a}}_i^s : \boldsymbol{a}_i \in \mathbb{R}^m, k \in \mathbb{R}\}$, is a vector space of dimension $\leq 1 + m$. This is because for every $j = 1, \ldots, m$, all monomials that contain a variable $\boldsymbol{u}_s[j]$ for some $s$ have the same coefficient (equal to $\boldsymbol{a}_i[j]$ for some $1 \leq i \leq n$). Explicit spanning sets are given by the following recursion. For each $j = 1, \ldots, m$ define $\widetilde{\boldsymbol{u}}_1[j], \ldots, \widetilde{\boldsymbol{u}}_w[j]$ recursively as

$$\widetilde{\boldsymbol{u}}_1[j] = \boldsymbol{u}_1[j]$$

$$\widetilde{\boldsymbol{u}}_s[j] = \boldsymbol{u}_s[j] + \sum_{\ell=1}^{s-1} \boldsymbol{u}_s[m+\ell]\widetilde{\boldsymbol{u}}_\ell[j]$$

for $s = 2, \ldots, w$. Then, $\{k + \boldsymbol{u}_s^T \widetilde{\boldsymbol{a}}_i^s : \boldsymbol{a}_i \in \mathbb{R}^m, k \in \mathbb{R}\}$ is contained in $\mathrm{span}\{1, \widetilde{\boldsymbol{u}}_s[1], \ldots, \widetilde{\boldsymbol{u}}_s[m]\}$. It follows that

$$\dim\left(\bigcup_{s=1}^{w} \{k + \boldsymbol{u}_s^T \widetilde{\boldsymbol{a}}_i^s : \boldsymbol{a}_i \in \mathbb{R}^m, k \in \mathbb{R}\}\right) \leq 1 + mw.$$

The dual space thus also has dimension $\leq 1 + mw$. The VC dimension of the family of $0/1$ classifiers induced by a finite-dimensional vector space of functions is at most the dimension of the vector space. Thus, the VC dimension of the set of classifiers induced by the dual space is $\leq 1 + mw$. Finally, applying the main result of Balcan et al. [8] in conjunction with Lemma 3.4 gives the desired pseudo-dimension bound. $\qquad\square$

*Proof of Theorem 3.6.* Applying cuts $\boldsymbol{u}^1, \ldots, \boldsymbol{u}^k \in [0,1]^m$ simultaneously is equivalent to sequentially applying the cuts

$$\boldsymbol{u}^1 \in [0,1]^m, \begin{bmatrix} \boldsymbol{u}^2 \\ 0 \end{bmatrix} \in [0,1]^{m+1}, \begin{bmatrix} \boldsymbol{u}^3 \\ 0 \\ 0 \end{bmatrix} \in [0,1]^{m+2}, \ldots, \begin{bmatrix} \boldsymbol{u}^k \\ 0 \\ \vdots \\ 0 \end{bmatrix} \in [0,1]^{m+k-1}.$$

Thus, the set in question is a subset of $\left\{f_{\boldsymbol{u}_1,\ldots,\boldsymbol{u}_{kw}} : \boldsymbol{u}_1 \in [0,1]^m, \ldots, \boldsymbol{u}_{kw} \in [0,1]^{m+kw-1}\right\}$ and has pseudo-dimension $O(mk^2w^2 \log(mkw(\alpha + \beta + n)))$ by Theorem 3.5. $\qquad\square$

### C.1 Data-dependent guarantees

The *empirical Rademacher complexity* [28] of a function class $\mathcal{F} \subseteq \mathbb{R}^{\mathcal{Y}}$ with respect to $y_1, \ldots, y_N \in \mathcal{Y}$ is the quantity

$$\mathcal{R}_{\mathcal{F}}(N; y_1, \ldots, y_N) = \mathbb{E}_{\sigma \sim \{-1,1\}^N}\left[\sup_{f \in \mathcal{F}} \frac{1}{N} \sum_{i=1}^{N} \sigma_i f(y_i)\right].$$

The expected Rademacher complexity $\mathcal{R}_{\mathcal{F}}(N)$ of $\mathcal{F}$ with respect to a distribution $\mathcal{D}$ on $\mathcal{Y}$ is the quantity

$$\mathcal{R}_{\mathcal{F}}(N) = \mathbb{E}_{y_1,\ldots,y_N \sim \mathcal{D}}[\mathcal{R}_{\mathcal{F}}(N; y_1, \ldots, y_N)].$$

Rademacher complexity, like pseudo-dimension, is another measure of the intrinsic complexity of the function class $\mathcal{F}$. Roughly, it measures how well functions in $\mathcal{F}$ can correlate to random labels. The following uniform convergence guarantee in terms of Rademacher complexity is standard: Let $[-\kappa, \kappa]$ be the range of the functions in $\mathcal{F}$. Then, for all distributions $\mathcal{D}$ on $\mathcal{Y}$, with probability at least $1 - \delta$ over the draw of $y_1, \ldots, y_N \sim \mathcal{D}$, for all $f \in \mathcal{F}$, $\mathbb{E}_{y \sim \mathcal{D}}[f(y)] - \frac{1}{N}\sum_{i=1}^{N} f(y_i) \leq 2\mathcal{R}_{\mathcal{F}}(N) + \kappa\sqrt{\frac{\ln(1/\delta)}{N}}$.

The following result bounds the Rademacher complexity of the class of tree-size functions corresponding to $w$ waves of $k$ CG cuts. The resulting generalization guarantee is more refined than the pseudo-dimension bounds in the main body of the paper. It is in terms of distribution-dependent quantities, and unlike the pseudo-dimension-based guarantees requires no boundedness assumptions on the distributions's support.

**Theorem C.1.** *Let $\mathcal{D}$ be a distribution over integer programs $(\boldsymbol{c}, A, \boldsymbol{b})$. Let*

$$\alpha_N = \mathop{\mathbb{E}}_{A_1,\ldots,A_N \sim \mathcal{D}} \left[ \max_{1 \leq i \leq N} \|A_i\|_{1,1} \right] \quad \text{and} \quad \beta_N = \mathop{\mathbb{E}}_{\boldsymbol{b}_1,\ldots,\boldsymbol{b}_N \sim \mathcal{D}} \left[ \max_{1 \leq i \leq N} \|\boldsymbol{b}\|_1 \right].$$

*The expected Rademacher complexity $\mathcal{R}(N)$ of the class of tree-size functions corresponding to $w$ waves of $k$ Chvátal-Gomory cuts with respect to $\mathcal{D}$ satisfies*

$$\mathcal{R}(N) \leq O\left( \kappa \sqrt{\frac{mk^2w^2\log(mkw(\alpha_N + \beta_N + n))}{N}} \right)$$

*where $\kappa$ is a cap on the size of the tree B&C is allowed to build.*

*Proof of Theorem C.1.* Let $\mathcal{F}_{\alpha,\beta}$ denote the class of tree-size functions corresponding to $w$ waves of $k$ CG cuts defined on the domain of integer programs with $\|A\|_{1,1} \leq \alpha$ and $\|\boldsymbol{b}\|_1 \leq \beta$, and let $\mathcal{F}$ denote the same class of functions without any restrictions on the domain. Applying a classical result due to Dudley [14], the empirical Rademacher complexity of $\mathcal{F}$ with respect to $(\boldsymbol{c}_1, A, \boldsymbol{b}), \ldots, (\boldsymbol{c}_N, A, \boldsymbol{b}_N)$ satisfies the bound

$$\mathcal{R}_{\mathcal{F}}(N; (\boldsymbol{c}_1, A, \boldsymbol{b}_1), \ldots, (\boldsymbol{c}_N, A, \boldsymbol{b}_N)) \leq 60\kappa \sqrt{\frac{\text{Pdim}\left(\mathcal{F}_{\max_i\|A_i\|_{1,1}, \max_i\|\boldsymbol{b}_i\|_1}\right)}{N}}.$$

Here, $\kappa$ is a bound on the tree-size function as is common in the algorithm configuration literature [5, 26, 27]. Taking expectation over the sample, we get

$$\mathcal{R}_{\mathcal{F}}(N) \leq 60\kappa \sqrt{\frac{\mathbb{E}\left[\text{Pdim}\left(\mathcal{F}_{\max_i\|A_i\|_{1,1}, \max_i\|\boldsymbol{b}\|_{1,1}}\right)\right]}{N}}$$

$$\leq 60\kappa \sqrt{\frac{\mathbb{E}\left[mk^2w^2\log(mkw(\max_i\|A_i\|_{1,1} + \max_i\|\boldsymbol{b}\|_1 + n))\right]}{N}}$$

$$\leq 60\kappa \sqrt{\frac{mk^2w^2\log(mkw(\alpha_N + \beta_N + n))}{N}}$$

by Theorem 3.6 and Jensen's inequality. $\qquad\square$

## D  Omitted proofs from Section 5

*Proof of Theorem 5.2.* Fix an arbitrary problem instance $x$. In Claim D.1, we prove that for any sequence of actions $\sigma \in \left(\times_{j=1}^t [T_j]\right)^\kappa$, there is a set of at most $\kappa \sum_{j=1}^t T_j^2$ halfspaces in $\mathbb{R}^d$ such that Algorithm 1 when parameterized by $\boldsymbol{\mu} \in \mathbb{R}^d$ will follow the action sequence $\sigma$ if and only if $\boldsymbol{\mu}$ lies in the intersection of those halfspaces. Let $\mathcal{H}_\sigma$ be the set of hyperplanes corresponding to those halfspaces, and let $\mathcal{H} = \bigcup_\sigma \mathcal{H}_\sigma$. Since there are at most $\prod_{j=1}^t T_j^\kappa$ action sequences in $\left(\times_{j=1}^t [T_j]\right)^\kappa$, we know that $|\mathcal{H}| \leq \kappa \left(\prod_{j=1}^t T_j^\kappa\right) \sum_{j=1}^t T_j^2$. Moreover, by definition of these halfspaces, we know that for any connected component $C$ of $\mathbb{R}^d \setminus \mathcal{H}$, across all $\boldsymbol{\mu} \in C$, the sequence of actions Algorithm 1 follows is invariant. Since the state transitions are deterministic functions of the algorithm's actions, this means that the algorithm's final state is also invariant across all $\boldsymbol{\mu} \in C$. Since the utility function is final-state-constant, this means that $f_{\boldsymbol{\mu}}(x)$ is constant across all $\boldsymbol{\mu} \in C$. Therefore, the sample complexity guarantee follows from Balcan et al. [8]. $\qquad\square$

**Claim D.1.** *Let $\sigma \in \left(\times_{j=1}^t [T_j]\right)^\kappa$ be an arbitrary sequence of actions. There are at most $\kappa \sum_{j=1}^t T_j^2$ halfspaces in $\mathbb{R}^d$ such that Algorithm 1 when parameterized by $\boldsymbol{\mu} \in \mathbb{R}^d$ will follow the action sequence $\sigma$ if and only if $\boldsymbol{\mu}$ lies in the intersection of those halfspaces.*

*Proof.* For each type of action $j \in [t]$, let $k_{j,1}, \ldots, k_{j,\kappa} \in [T_j]$ be the sequence of action indices taken over all $\kappa$ rounds. We will prove the claim by induction on the step of B&C. Let $\mathcal{T}_\tau$ be the state of the B&C tree after $\tau$ steps. For ease of notation, let $\overline{T} = \sum_{j=1}^t T_j^2$ be the total number of possible actions squared.

**Induction hypothesis.** For a given step $\tau \in [\kappa t]$, let $\kappa_0 \in [\kappa]$ be the index of the current round and $t_0 \in [t]$ be the index of the current action. There are at most $(\kappa_0 - 1)\overline{T} + \sum_{j=1}^{t_0} T_j^2$ halfspaces in $\mathbb{R}^d$ such that B&C using the scoring rules $\sum_{i=1}^{d_j} \mu_j[i]\texttt{score}_{j,i}$ for each action $j \in [t]$ builds the partial search tree $\mathcal{T}_\tau$ if and only if $(\boldsymbol{\mu}_1, \ldots, \boldsymbol{\mu}_t) \in \mathbb{R}^d$ lies in the intersection of those halfspaces.

**Base case.** In the base case, before the first iteration, the set of parameters that will produce the partial search tree consisting of just the root is the entire set of parameters, which vacuously is the intersection of zero hyperplanes.

**Inductive step.** For a given step $\tau \in [\kappa t]$, let $\kappa_0 \in [\kappa]$ be the index of the current round and $t_0 \in [t]$ be the index of the current action. Let $s_\tau$ be the state of B&C at the end of step $\tau$. By the inductive hypothesis, we know that there exists a set $\mathcal{H}$ of at most $(\kappa_0 - 1)\overline{T} + \sum_{j=1}^{t_0} T_j^2$ halfspaces such that B&C using the scoring rules $\sum_{i=1}^{d_j} \mu_j[i]\texttt{score}_{j,i}$ for each action $j \in [t]$ will be in state $s_\tau$ if and only if $(\boldsymbol{\mu}_1, \ldots, \boldsymbol{\mu}_t) \in \mathbb{R}^d$ lies in the intersection of those halfspaces. Let $\kappa_0' \in [\kappa]$ be the index of the round in step $\tau + 1$ and $t_0' \in [t]$ be the index of the action in step $\tau + 1$, so

$$(\kappa_0', t_0') = \begin{cases} (\kappa_0, t_0 + 1) & \text{if } t_0 < t \\ (\kappa_0 + 1, 1) & \text{if } t_0 = t. \end{cases}$$

We know B&C will choose the action $k^* \in [T_{t_0'}]$ if and only if

$$\sum_{i=1}^{d_{t_0'}} \mu_{t_0'}[i]\texttt{score}_{t_0',i}(k^*, s_\tau) > \max_{k \neq k^*} \sum_{i=1}^{d_{t_0'}} \mu_{t_0'}[i]\texttt{score}_{t_0',i}(k, s_\tau).$$

Since these functions are linear in $\boldsymbol{\mu}_{t_0'}$, there are at most $T_{t_0'}^2$ halfspaces defining the region where $k_{t_0', \kappa_0'} = \text{argmax} \sum_{i=1}^{d_{t_0'}} \mu_{t_0'}[i]\texttt{score}_{t_0',i}(k, s_\tau)$. Let $\mathcal{H}'$ be this set of halfspaces. B&C using the scoring rule $\sum_{i=1}^{d_{t_0'}} \mu_{t_0'}[i]\texttt{score}_{t_0',i}$ arrives at state $s_{\tau+1}$ after $\tau + 1$ iterations if and only if $\boldsymbol{\mu}_{t_0'}$ lies in the intersection of the $(\kappa_0' - 1)\overline{T} + \sum_{j=1}^{t_0'} T_j^2$ halfspaces in the set $\mathcal{H} \cup \mathcal{H}'$. $\qquad\square$