# OpenReview forum: "Sample Complexity of Tree Search Configuration: Cutting Planes and Beyond"
_NeurIPS.cc/2021/Conference — NeurIPS 2021 Spotlight_

### Official Review · Reviewer_4fY3 · 2021-07-13

**Rating:** 7
**Confidence:** 2

**Summary:**

This paper studies the sample complexity of cutting plane selection in branch-and-cut algorithms for integer programming problems. The authors study a handful of different settings, with the goal of characterizing the intrinsic difficulty of learning: (i) the tree size upon adding a single cut at the root, (ii) the tree size upon adding a sequence of cuts at the root, (iii) the tree size upon adding a series of waves of cuts at the root, and (iv) the score of cuts according to a set of predefined scoring rules. Finally, the authors generalize the results to bound the sample complexity of generic tree search policies with sequential operations that can be scored.

**Main Review:**

The paper is nicely written, tackles an interesting problem, and covers a nice range of related, but still distinct, settings. I view it as a worthy addition to the literature, though I am not particularly well-suited to evaluate the learning aspects of the work (i.e. its novelty or deepness).

I have two questions/comments for the authors:
1.This is nitpicky, but: [10], as cited on L136, restricts attention to bounded polyhedron, whereas the authors do not make the same assumption.
2. How much of this machinery is specialized for integer programming as opposed to mixed-integer programming? I would be interested to hear the authors thoughts, though I do not feel that a lengthy discussion is needed in the paper itself.

**Time Spent Reviewing:**

1

---

> ### Author Response · Authors · 2021-08-10
> **Response to Reviewer 4fY3**
>
> Thank you for the review and the thoughtful feedback! We appreciate your evaluation that the paper tackles an interesting problem and covers a range of different settings.
>
> We respond to the questions in the review below.
> > This is nitpicky, but: [10], as cited on L136, restricts attention to bounded polyhedron, whereas the authors do not make the same assumption.
>
> This result actually holds for any polyhedra of the form Ax <= b, for A and b integral. A proof is given in Lemma 5.13 of the textbook “Integer Programming” by Conforti, Cornuejols, and Zambelli. Thank you for pointing out that the paper by Chvatal only dealt with bounded polyhedra. We will include this updated reference.
>
> > How much of this machinery is specialized for integer programming as opposed to mixed-integer programming?
>
> The results in Section 3 on Chvatal-Gomory cuts are restricted to integer programs (since Chvatal-Gomory cuts are valid only for pure integer programs), but the results in Section 4 on cut selection policies and Section 5 on generic tree search apply to mixed-integer programming as well.

---

### Official Review · Reviewer_i38d · 2021-07-14

**Rating:** 7
**Confidence:** 4

**Summary:**

The paper focuses on the sample complexity of learning to select Chvatal-Gomory cuts for integer linear programming. We assume that there is an unknown distribution that generates ILP instances. CG cuts are parametrized by a set of weights, one per constraint. How large should the set of training instances be for one to accurately estimate the "goodness" of a given parametrization? This is the main question tackled here.

Using the data-driven algorithm design framework of Balcan et al. [8], this paper shows that three flavors of the learning problem can be analyzed effectively. The main contribution is to show that there is structure to the cut generation process as its parameters vary; the space of possible cuts can be partitioned, the form of the boundaries that determine the partition is identified, and the behavior of the cut generation is constant within each region. These can be plugged into a very general PAC learning bound from Balcan et al. [8].

Additionally, the sample complexity of generic tree search is analyzed. It is shown that variable, node, and cut selection can be parameterized simultaneously, each with its own additive scoring function, and sample complexity bounds can be derived accordingly. This result generalizes a previous branching-only bound from Balcan et al. [5].


**Main Review:**

Clarity: The paper is very well-written and easy to read. Proofs are either concise or sketched in the main text, which I appreciate.

Originality: The sample complexity of learning to generate CG cuts is a new problem as far as I know. Deriving the partitions of the parameter spaces do involve some careful analysis.

Quality: The paper is expertly executed. I have checked the proofs to the extent that I am capable, and identified one potential mistake which is likely inconsequential:

In line 203, you say $\sum_{i=1}^{n}{2A_i} \in O(\lVert A \rVert_{1,1})$. I suspect this should be $O(n\lVert A \rVert_{1,1})$: consider the case where all column norms are equal, then summing them up gives $2n \lVert A \rVert_{1,1}$.

If you ignore logarithmic factors in the sample complexity bound of line 210, this missing factor of $n$ is ignored anyways, so even if I were right about this, it is maybe not a big issue. I'd like the authors to comment on this.

Significance: There's a basic issue with the set of results in section 3. The values of the parameter vector depend on the index of the constraint, i.e., $u[1]$ corresponds to the first constraint, etc. Now consider two identical ILP instances with the constraints permuted. The same parameter vector can potentially produce a different cut. A practical scenario for this is a graph optimization problem, say Max. Independent Set (MIS), where each linear constraint expresses independence along an edge of the graph; there is no intrinsic ordering to the edges and thus to the constraints. This is to say that unlike the scoring rules in section 4, the parameters in section 3 make little sense unless you make strong assumptions on the distribution of instances, e.g., if all instances have the exact same constraints but a random objective function, which makes the ordering argument irrelevant.

I am not 100% sure of it, but I think section 4 does not suffer from this issue, as the scoring rules take pre-generated cuts as input.

More broadly though, while the technical work involved in deriving the sample complexity bounds in this paper is solid, I don't see what the theory tells us about the practice: the sample complexity bounds often require a large number of instances (e.g., line 210, quadratic in the maximum search tree size), and there is no clear path to a practical empirical risk minimization procedure that exploits the partition of the parameter space. I would be interested in hearing what the authors think about this.

Minor:
- Line 318: "which allow the cuts B&C that selects" --> "which allow the cuts that B&C selects"
- This paper might be relevant to your literature review: Baltean-Lugojan, Radu, et al. Scoring positive semidefinite cutting planes for quadratic optimization via trained neural networks. Working paper., 2019, http://www. optimization-online. org/DB_HTML/2018/11/6943. html, 2019.


**Time Spent Reviewing:**

8

---

> ### Author Response · Authors · 2021-08-10
> **Response to Reviewer i38d**
>
> Thank you for the review and the thoughtful feedback! Thank you also for recognizing the novelty of the problem. We are glad to hear that you found the paper to be expertly executed.
>
> We respond to the questions in the review below.
>
> > In line 203, you say $\sum_{i = 1}^n 2A_i \in O(||A||\_{1,1})$. I suspect this should be $O(n||A||\_{1,1})$: consider the case where all column norms are equal, then summing them up gives $2n||A||_{1,1}$. If you ignore logarithmic factors in the sample complexity bound of line 210, this missing factor of $n$ is ignored anyways, so even if I were right about this, it is maybe not a big issue. I'd like the authors to comment on this.
>
> We are using $||A||\_{1, 1}$ to denote the sum of the absolute values of all the entries in $A$ (not the 1-norm $||A||\_1$ which is the maximum column norm), so $\sum_{i=1}^n A_i = ||A||_{1, 1}$. We will clarify this in Section 2.
>
> > There's a basic issue with the set of results in section 3. The values of the parameter vector depend on the index of the constraint, i.e., u[1] corresponds to the first constraint, etc. Now consider two identical ILP instances with the constraints permuted. The same parameter vector can potentially produce a different cut [...] This is to say that unlike the scoring rules in section 4, the parameters in section 3 make little sense unless you make strong assumptions on the distribution of instances, e.g., if all instances have the exact same constraints but a random objective function, which makes the ordering argument irrelevant.
>
> It is standard that in integer programming solvers, the steps that the solver takes (and thus the speed) depend on the order in which constraints are listed in the input model, and that same issue applies to our work as well. However, integer programs are typically automatically generated and the generation code typically generates constraints in the same order for all instances. More formally, if we fix an ordering over the variables, we can simply assume without loss of generality that the constraints and cuts are ordered lexicographically, so the constraints cannot be permuted across instances. (In a bit more detail, given constraints with coefficients $[a_1, …, a_n, b]$ and $[a’_1, …, a’_n, b’]$, the first constraint would come first in the lexicographic order if $a_1 > a’_1$ and second if $a_1 < a’_1$. If $a_1 = a’_1$ it would come first in the order if $a_2 > a’_2$ and second if $a_2 < a’_2$, and so on.) Moving on to Section 4, one could even view our work as mitigating the issue of unfortunate constraint ordering in the input model because that is partially addressed by well-tuned scoring rules.
>
> Moreover, in many (if not most) IP settings, similar IPs are being solved and there can be good cuts that carry across instances--for example, in applications where the constraints stay the same or roughly the same across instances, and only the objective changes. One high-stakes  example of this is the feasibility checking problem in the billion-dollar incentive auction for radio spectrum, where prices change but the radiowave interference constraints do not change.
> We will add this discussion to Section 2.
>
> > I don't see what the theory tells us about the practice: the sample complexity bounds often require a large number of instances (e.g., line 210, quadratic in the maximum search tree size), and there is no clear path to a practical empirical risk minimization procedure that exploits the partition of the parameter space.
>
> Sample complexity bounds are important because if the parameterized class of cuts or cut-selection policies that we optimize over is highly complex and the training set is too small, the learned cut or cut-selection policy might have great average empirical performance over the training set but terrible future performance. In other words, the parameter configuration procedure may overfit to the training set.
>
> Indeed, tree size is an extremely complicated, volatile function of the cut parameters, with many jump discontinuities: the addition of even a single cut at the root can impact many different aspects of B&C, even the variables that are branched on at the bottom of the tree. We illustrate this volatility in Figure 1 and Lemma 3.1.  Therefore, it’s not clear a priori whether it’s even possible to avoid overfitting. Surprisingly, we show that there is structure governing the volatile tree-size function which allows us to provide positive sample complexity guarantees.
>
> Moreover, we emphasize that the bounds we provide are uniform-convergence: we prove that given enough samples, uniformly across all parameter settings, the difference between average and empirical performance is small. In other words, these bounds hold for any procedure one might use to optimize over the training set: manual or automated, optimal or suboptimal. No matter what parameter setting the configuration procedure comes up with, the user can be guaranteed that so long as that parameter setting has good average empirical performance over the training set, it will also have strong future performance.
>
> A strength of our guarantees is that they hold for any distribution over problem instances, and thus classic results from learning theory tell us that our bounds unavoidably must depend on the range of the performance function ($\kappa$ in our case). The tree size bound could be estimated from a small amount of data, as in research by Balcan, Sandholm, and Vitercik [ICML’20]. Another approach is to provide data-dependent sample complexity bounds, rather than worst-case sample complexity bounds, and this is a great direction for future research that we will add to Section 6.

---

> > ### Comment · Reviewer_i38d · 2021-09-01
> > **Thank you**
> >
> > I have increased my rating from 6 to 7 following the rebuttal. I appreciate the effort. If the paper is accepted, I hope that the discussion about the ordering is included in the appendix. I agree that uniform convergence results are great to have and a key contribution here, but because ERM is not addressed in this work and MIP solving is a very active *empirical* domain, recognizing that uniform convergence bounds may not be practically useful is important for the sake of transparency.

---

### Official Review · Reviewer_U5ky · 2021-07-15

**Rating:** 7
**Confidence:** 4

**Summary:**

Combinatorial optimization solvers often add cutting planes during solving, which are additional constraints in the problem that accelerate solving. These cutting planes have a major impact on the size of the final branch-and-bound tree, and therefore of the speed of the overall solving, but their impact is badly understood theoretically. At minimum, it is empirically understood that the size of the branch-and-bound tree is a complicated and sensitive function of the parameters of the cutting planes that were added during solving: a slightly different cutting plane might have been much more, or much less, effective.

In this paper the authors look at the problem of trying to learn these functions that relate the size of the branch-and-bound tree (a measure of solving performance) as a function of the instance data (A, b, c) when cuts with given parameters u_1, ..., u_k are added to the problem. They compute an order of magnitude of samples required to learn well this function, both for a single cut, a sequence of cuts, or rounds of sequences of cuts (as are used in practice). They also compute the number of samples required for the generalized problem of predicting tree size as a function of the instance data (A, b, c) for an algorithm that would pick the cut parameters to maximize a convex combination of scores, for given algorithm hyperparameters (the convex weights.) Finally, they do the same for more general algorithms that would pick an action, during solving, by maximizing a convex combination of scores, such as when doing node selection or variable selection.

**Limitations And Societal Impact:**

As I mentioned, although I like in general the contributions, I found the tone sometimes a bit too bombastic in the claim of its breath. On a positive note however, they do discuss how their results are mostly focused on Gomory fractional cuts and not other families.

**Main Review:**

I think the paper is overall interesting. Although branch-and-cut is the main algorithm used by combinatorial optimization solvers, very little theory has been developed for it, mostly because it is devilishly difficult. The current paper develops some theory related to a more tangential aspect (how hard is it to learn the function mapping cuts, or scoring weights, to the tree size?), but which might be useful for cut selection. In some sense only the initial results of the paper (Theorem 3.1, Lemma 3.2, Lemma 3.4) really seem to have to do with cutting planes per se; the rest is more a consequence of generic results about learning discrete functions, like the tree size, as a function of continuous parameters. Nonetheless the results are not trivial and are certainly original.

On the criticisms side, I think there could be ways for the presentation in Section 4 to be improved.
1) I think the authors should really separate better the paragraph starting at line 298 in two, at line 301. I found the lack of break confusing: the text switches from discussion Theorem 4.1 to moving on to the next result without a breath in between.
2) I think the authors should explain better in Theorem 4.1 what is the main result of Balcan et al. (2021) (Theorem 3.3?) and how it gets invoked in the proof, since the setup and notation of that second paper is quite different from this one. Right now I find the current proof a bit too hand-wavey for my taste, for what is ultimately (with Theorem 5.2) the most interesting result of the paper.
3) My third question is perhaps a bit more philosophical, but is there a good reason why the bound of Theorem 4.1 seems to have no relationship at all with the scoring functions? One might have thought that more complex scoring functions would have led to tree sizes that are more sensitive to their input, and therefore needing more samples to learn, but your bound seems independent of their complexity. Perhaps a little high level explanation would be welcome in the text.

Otherwise, I think the literature review section could be improved. There is a lot of applied papers missing, mostly from more recent years - the list is far from being up to date, both for variable selection, node selection and cut selection. For cut selection there is the recent Huang et al. (2021); for node selection there is Yilmaz and Yorke-Smith (2021). For variable selection (the most studied problem) there is quite a bit missing out, probably at least 10 papers or so since 2017-2018. Conversely, I am not sure I understand the relevance of the paragraph discussing Ferber et al. (2020), except that the method uses cutting planes - this has not really much to do with branch-and-cut?

I also think the tone is sometimes a bit too grandiose in the abstract and the paper. "In this paper we prove the first guarantees for learning high-performing cut-selection policies tailored to the instance distribution at hand using samples" makes it sound like the authors have a procedure with guarantees to produce high-quality cuts - this is not the case. Or in sentences like, "we bound the sample complexity of learning high-performing cutting planes" - I don't think they bound at all the sample complexity of learning the (optimal) planes, they bound the sample complexity of learning a metric, the tree size. This is different in my mind. Now, I do agree that the results of the paper might be useful for deriving such procedures-with-guarantees, for example by optimization of the empirical mean performance over instances (empirical risk minimization), but this is not discussed in the paper, and there would be more work needed to get there. In fact, I think the paper does in that respect an underwhelming job of explaining why these results are interesting, and on that aspect better explaining how they could be useful for cutting plane algorithms would be useful.

Another more minor comment: in Section 3 (and elsewhere in the paper), the authors discuss "waves of cuts". I am personally much more used to hear about "rounds" of cuts, which I think (perhaps incorrectly) is the more standard term in the combinatorial optimization community. To help readers, perhaps it would be valuable to switch to this terminology? Unless there is a distinction I did not quite grasp between the two concepts.

Overall, my recommendation would be acceptance of the paper, with the criticisms I described above having been addressed.

**Time Spent Reviewing:**

6

---

> ### Author Response · Authors · 2021-08-10
> **Response to Reviewer U5ky**
>
> Thank you for the review and the thoughtful feedback! We are glad to hear that you found the problem to be interesting and the results to be original.
>
> In the revision, we will make the suggested writing changes regarding the paragraph starting at line 298 and add the suggested references to Section 1.2. We respond to the questions in the review below.
>
> > I think the authors should explain better in Theorem 4.1 what is the main result of Balcan et al. (2021) (Theorem 3.3?) and how it gets invoked in the proof.
>
> At a high level, the main result of Balcan et al. (2021) is a general method for proving sample complexity bounds based on the structure exhibited by a combinatorial problem like integer programming. In our paper, the main challenge is uncovering structure exhibited by the branch-and-cut (B&C) tree size as a function of cuts or cut-selection policy parameters. In the revision, we will elucidate our use of the main result by Balcan et al.
>
> > My third question is perhaps a bit more philosophical, but is there a good reason why the bound of Theorem 4.1 seems to have no relationship at all with the scoring functions? One might have thought that more complex scoring functions would have led to tree sizes that are more sensitive to their input, and therefore needing more samples to learn, but your bound seems independent of their complexity.
>
> To answer this question, we first recall that Theorem 4.1 provides sample complexity bounds for learning a cut-selection policy that is defined as a convex combination of a finite set of base scoring rules. The tunable parameters define the convex combination. Essentially, the proof works by showing that for any fixed integer program, there’s a finite partition of the parameter space such that in any one region of the partition, the tree that B&C builds is fixed. At a high level, this allows us to understand the “intrinsic complexity” of B&C tree size as a function of the parameters.
>
> We illustrate why the bounds do not depend on the complexity of the scoring rules themselves with the following simple example. Suppose we are learning a convex combination between two scoring rules (1 - ρ) * score_1 + ρ * score_2. Moreover, suppose we will only add one of two cuts, c_1 or c_2, at the root, and no other cuts during B&C. Due to the linear dependence on the parameter ρ, there is a threshold ρ’ where:
> - For all ρ < ρ’, (1 - ρ) * score_1(c_1) + ρ * score_2(c_1) > (1 - ρ) * score_1(c_2) + ρ * score_2(c_2), and
> - For all ρ > ρ’, (1 - ρ) * score_1(c_1) + ρ * score_2(c_1) < (1 - ρ) * score_1(c_2) + ρ * score_2(c_2) (or vice versa).
>
> Therefore, for all ρ < ρ’, we will add the cut c_1 and for all ρ > ρ’, we will add the cut c_2. This is a highly simplified version of the argument we use to show that even in multiple dimensions, when adding cuts throughout the tree, there’s a partition of the parameter space where in any one region, the tree that B&C builds is fixed. This argument does not depend at all on the complexity of the scoring rules score_1 and score_2, and similarly the generalized argument does not depend on the complexity of the scoring rules either.
>
> It is possible that one could obtain even better bounds by taking into account the complexity of the scoring rules, and we will include this as a direction for future research in Section 6.
>
> > I think the paper does in that respect an underwhelming job of explaining why these results are interesting, and on that aspect better explaining how they could be useful for cutting plane algorithms would be useful.
>
> Sample complexity bounds are important because if the parameterized class of cuts or cut-selection policies that we optimize over is highly complex and the training set is too small, the learned cut or cut-selection policy might have great average empirical performance over the training set but terrible future performance. In other words, the parameter configuration procedure may overfit to the training set.
>
> Indeed, as the reviewer mentions, tree size is an extremely complicated, volatile function of the cut parameters, with many jump discontinuities: the addition of even a single cut at the root can impact many different aspects of B&C, even the variables that are branched on at the bottom of the tree. We illustrate this volatility in Figure 1 and Lemma 3.1. Therefore, it’s not clear a priori how a small change to the cut will impact the size of the tree that B&C builds, or whether it’s even possible to analyze this function. Nonetheless, understanding this function is crucial for proving sample complexity bounds. Surprisingly, we show that there is structure governing these volatile functions, which allows us to provide our guarantees.
>
> Moreover, we emphasize that the bounds we provide are uniform-convergence: we prove that given enough samples, uniformly across all parameter settings, the difference between average and empirical performance is small. In other words, these bounds hold for any procedure one might use to optimize over the training set: manual or automated, optimal or suboptimal. No matter what parameter setting the configuration procedure comes up with, the user can be guaranteed that so long as that parameter setting has good average empirical performance over the training set, it will also have strong future performance.
>
> We will incorporate this discussion into Section 1.1 of the revision.
>
> > I also think the tone is sometimes a bit too grandiose
>
> Thank you for pointing out that the tone might be perceived as grandiose. We will make the following revisions according to the reviewer’s recommendations:
> - We will change "In this paper we prove the first guarantees for learning high-performing cut-selection policies tailored to the instance distribution at hand using samples" to "In this paper we provide sample complexity bounds for cut-selection in branch-and-cut (B&C). Given a training set of integer programs sampled from an application-specific input distribution and a family of cut selection policies, these guarantees bound the number of samples sufficient to ensure that using any policy in the family, the size of the tree B&C builds on average over the training set is close to the expected size of the tree B&C builds."
> - We will change “As our first main contribution, we bound the sample complexity of learning high-performing cutting planes. Fixing a family of cutting planes, these guarantees bound the number of samples sufficient to ensure that for any sequence of cutting planes from the family, its average performance over the samples is close to its expected performance" to "As our first main contribution, we provide sample complexity bounds of the following form: fixing a family of cutting planes, we bound the number of samples sufficient to ensure that for any sequence of cutting planes from the family, the average size of the B&C tree is close to the expected size of the B&C tree."
>
> We will also take a careful pass to revise any other such sentences.

---

### Official Review · Reviewer_D3H9 · 2021-07-16

**Rating:** 7
**Confidence:** 3

**Summary:**

Linear integer programming is an incredibly important algorithmic tool that is used widely in practice.  In general the problem is NP-hard and so we have to resort to heuristics. These heuristics often use the branch-and-cut framework: we branch on "guesses" on variables and we cut by adding valid inequalities that tightens the linear relaxation of the integer program.

When dealing with such heuristics, an obvious question arises: which cut should I add and on which variable should I branch? The goal here is to limit  the size of the search tree and therefore getting a good running time. There has been a growing amount of experimental work on using ML algorithms to guide this choice.

The paper under submission considers this problem from a more theoretical perspective. The main results bound the sample complexity for finding "good" cuts complementing and generalizing a prior work that studied this question for variable selection.

**Ethical Concerns:**

No ethical concerns.

**Limitations And Societal Impact:**

I do wish the authors would have discussed more why the sample complexity is important for cut selection. Perhaps it is a necessity but it is not sufficient (the sample complexity of exponentially many functions is low but we can't try all). As aforementioned, it would also have been good if the authors would have been given a concrete examples where we would expect to have a distribution of IPs and select our cuts depending on this distribution and not the instance.


**Main Review:**

The paper studies an important problem and give original new contributions.  The paper is also written extremely well and is a pleasure to read.  In what follows, I detail some of the contributions of the paper and comment on their strengths and weaknesses.

The first question that the authors study is the following. Suppose you have an (unknown) distribution over m-constraint IPs. You would like to find the best Chvatal-Gomory (CG) cut u in the following sense. If I add u to the IP then I would like to have the smallest search tree in expectation over the randomly chosen IP (from the distribution). More specifically, the authors study the sample complexity of u and show that it is near linear in m under reasonable norm bounds on the coefficients of the IP instances.  The proofs of this is not hard and uses a connection to pseudo-dimension used in prior work. The key insight is the observation that for a fixed IP the number of Chvatal-Gomory cuts of interest is  exponential.  The authors also generalize this to when you add several cuts to the root of the search tree.

The  strength of the first line of results is that it give new insights to a classic and very natural family of cuts (Chvatal-Gomory cuts). The weaknesses are that (1) the results do not tell us much about when we can find a good cut efficiently; and, as the author points out, (2) the current cut does not depend on the actual IP only on the distribution which is quite unnatural.  It would be good if the authors further explain when this setting is natural and also to comment on the efficiency issue.

Indeed, many solvers uses different scoring rules (that depend on the IP) to select the cuts to add. This is the second part of the paper where the authors analyze the sample complexity of learning such scores. The statement of this result is interesting. The techniques to get it are not striking: basically they show that a good combination of d scoring rules is learnable. Assuming the coefficients in front of these scoring rules don't have huge bit complexity, this follows since then the number of options is say poly(n)^d (or even exp(n)^d if we allow polynomial bit complexity in our coefficients). Now the authors have to work  a little more since they don't make such an assumption.

The authors also generalize this to a more generalize tree search model in which they are able to recover and generalize prior work that only considered the sample complexity of variable selection.

Overall, I see this as a nice paper that is interesting. It touches on a very interesting problem where we can expect ML to lead to huge speedups. It is unclear how the the sample complexity of the problems considered here will inform such improvements and it would be great if the authors would comment more on this. Overall, I'd recommend the paper to be accepted.



**Time Spent Reviewing:**

5

---

> ### Author Response · Authors · 2021-08-10
> **Response to Reviewer D3H9**
>
> Thank you for the review and the thoughtful feedback! We appreciate your evaluation that the paper touches on an interesting problem where we can expect ML to lead to huge speedups, and that it provides original new contributions.
>
> We respond to the questions in the review below.
>
> > the current cut does not depend on the actual IP only on the distribution […] it would also have been good if the authors would have been given a concrete examples where we would expect to have a distribution of IPs and select our cuts depending on this distribution and not the instance.
>
> First, as the reviewer points out, Sections 4 and 5 provide sample complexity bounds for optimizing cut selection policies. In this case, the pool of cuts we choose from (e.g., the set of all Gomory cuts, generated in the normal instance-specific way) and the cuts selected by the policy will depend on the input IP.
>
> Moreover, in many (if not most) IP settings, similar IPs are being solved and there can be good cuts that carry across instances--for example, in applications where the constraints stay the same or roughly the same across instances, and only the objective changes. One high-stakes  example of this is the feasibility checking problem in the billion-dollar incentive auction for radio spectrum, where prices change but the radiowave interference constraints do not change. We will add this discussion to Section 2.
>
> > [The paper] touches on a very interesting problem where we can expect ML to lead to huge speedups. It is unclear how the sample complexity of the problems considered here will inform such improvements.
>
> Sample complexity bounds are important because if the parameterized class of cuts or cut-selection policies that we optimize over is highly complex and the training set is too small, the learned cut or cut-selection policy might have great average empirical performance over the training set but terrible future performance. In other words, the parameter configuration procedure may overfit to the training set.
>
> Indeed, tree size is an extremely complicated, volatile function of the cut parameters with many jump discontinuities: the addition of even a single cut at the root can impact many different aspects of B&C, even the variables that are branched on at the bottom of the tree. We illustrate this volatility in Figure 1 and Lemma 3.1. Therefore, it’s not clear a priori whether it’s even possible to avoid overfitting. Surprisingly, we show that there is structure governing the volatile tree-size function which allows us to provide positive sample complexity guarantees.
>
> Moreover, we emphasize that the bounds we provide are uniform-convergence: we prove that given enough samples, uniformly across all parameter settings, the difference between average and empirical performance is small. In other words, these bounds hold for any procedure one might use to optimize over the training set: manual or automated, optimal or suboptimal. No matter what parameter setting the configuration procedure comes up with, the user can be guaranteed that so long as that parameter setting has good average empirical performance over the training set, it will also have strong future performance.
>
> We will incorporate this discussion into Section 1.1 of the revision.

---

### Decision · Program_Chairs · 2021-09-27

**Decision:**

Accept (Spotlight)

**Comment:**

The paper shows sampling bounds for the problem of identifying good cuts in branch-and-bound search using cutting planes. The reviewers appreciated the importance of the problem and the contribution of the paper. Some reviewers were concerned that sampling bounds by themselves do not provide efficient algorithms for finding good cuts, but these concerns were mostly resolved during the rebuttal phase.